# Optimizing LSTM networks and feature selection algorithms using GEE data

**Mohammad Kazemi[1], Reza Naderi Samani[2], Narges Kariminejad[3]***

1 Hormoz Studies and Research Center, University of Hormozgan, Bandar Abbas, Iran, 2 Researcher, Isfahan Agricultural and Natural Resources Research Center, Agricultural Research and Education Organization (AREEO), Isfahan, Iran, 3 Department of Natural Resources and Environmental Engineering, College of Agriculture, Shiraz University, Shiraz, Iran

* narges.karimi991@gmail.com

## Abstract

Feature selection uncertainty and suboptimal model configuration in flood susceptibility mapping (FSM) are critical challenges for disaster risk reduction. This study introduces a novel integrated framework that couples feature selection strategy with metaheuristic-optimized deep learning for high-precision FSM in the flood-prone Khuzestan Province, Iran. An initial set of 19 factors was sourced from Google Earth Engine (GEE). The most influential variables were identified using an ensemble of nine feature selection methods, including Boruta, Boruta-SHAP, Elastic-Net, Mutual Information, Permutation Importance, Recursive Feature Elimination (RFE), Sequential Forward Selection (SFS), Stability Selection, and Deep Feature Importance. For model development, 1,000 sample points were used, consisting of 500 randomly selected non-flood points (value 0) and 500 flood points (value 1), with the trained model subsequently generalized to the entire study area. In this process, a frequency-based consensus rule was applied, whereby variables were retained only if selected by a majority of methods. This process established the Normalized Difference Vegetation Index (NDVI) and Daily Minimum Temperature (TMMN) as the most critical predictors. A Long Short-Term Memory (LSTM) was developed using this optimal feature set and subsequently enhanced through hyperparameter optimization with five advanced metaheuristic algorithms, including WOA, GWO, OOA, CSA, and HOA. The model validation demonstrated that optimization significantly boosted performance, with the LSTM-WOA model emerging as superior, achieving the highest F1-Score (0.88) and Cohen's Kappa (0.75). The final FSM identified the northwestern and central regions as the highest susceptibility. The study innovation lies in its formalized consensus feature selection and comparative metaheuristic optimization, providing a reliable tool for FSM in arid and semi-arid regions.

**Data availability statement:** All relevant data are within the manuscript and its Supporting Information files.

**Funding:** The author(s) received no specific funding for this work.

**Competing interests:** NO authors have competing interests.

# 1. Introduction

Floods are one of the most destructive natural disasters worldwide, and their frequency and intensity are increasing due to climate change and growing human activity in vulnerable areas [1,2]. To reduce flood risks proactively, it is essential to accurately identify areas that are prone to flooding. FSM provides this information by identifying areas prone to flooding based on the natural characteristics of the landscape [2].

Over time, FSM methods have evolved from simple, expert-based approaches to advanced quantitative models. Recently, the use of Artificial Intelligence (AI), especially Machine Learning (ML) and Deep Learning (DL), combined with rich geospatial data from platforms like GEE, has transformed how we predict environmental hazards [3]. Models such as Random Forest (RF) and Support Vector Machines (SVM) have proven effective at capturing complex, non-linear relationships between flood-related factors and flood occurrence [4,5]. However, flooding often depends on past conditions, which means the process has a strong temporal dimension. This is where Long Short-Term Memory (LSTM) networks, a powerful class of deep learning models, are employed. Their inherent gating mechanisms enable learning from sequential data and capturing long-term dependencies, making them particularly well-suited for flood forecasting [6–8].

Despite their potential, using ML/LSTM models effectively for FSM faces two major and related challenges. The first is input feature selection. How well a data-driven model performs depends heavily on choosing the right set of input variables [9]. Although many feature selection methods exist (e.g., filter, wrapper, and embedded approaches [4,5,10]), different methods often pick different sets of "important" variables, creating uncertainty [11]. What's missing is a formal, consensus-based strategy that combines results from multiple selection methods to arrive at a stable and reliable set of predictors. This gap is particularly noticeable in hydrological FSM research [12].

The second challenge is hyperparameter tuning. The performance of an LSTM model is highly sensitive to choices like the number of layers, hidden units, and learning rate [13]. Metaheuristic algorithms, which are inspired by natural phenomena, have proven to be very effective at automating this tuning process, exploring complex search spaces far more efficiently than grid or random search. Studies have successfully combined models with algorithms such as the Genetic Algorithm (GA), Grey Wolf Optimizer (GWO), and Whale Optimization Algorithm (WOA) [14–16]. However, most existing work relies on just one or two optimizers [17]. A systematic comparison of metaheuristic algorithms for optimizing LSTM-based FSM models is still lacking, particularly one that includes both well-established and more recent algorithms, such as OOA and HOA. As a result, their relative performance in this setting remains largely untested [18,19].

In short, three key research gaps emerge: (1) no systematic benchmarking of multiple metaheuristic algorithms for optimizing LSTM networks in FSM; (2) no formal consensus mechanism for combining results from different feature selection methods to obtain robust input variables; and (3) no integrated framework that brings

consensus-based feature selection and comprehensively optimized LSTM together. Addressing this last gap is especially important, as it introduces a dual-layer optimization approach—refining both the model's inputs and its internal structure—something rarely seen in standard DL applications [13].

This study focuses on Khuzestan Province in southwestern Iran, a region that experiences severe and repeated flooding despite its arid to semi-arid climate [20]. Its vast, flat plains, part of the Mesopotamian alluvial basin and crossed by major rivers like the Karun and Karkheh, are constantly at risk due to a combination of very low topographic relief, highly variable seasonal rainfall, and intensive agriculture. This unique mix of natural and human-driven factors makes Khuzestan both an ideal and a demanding setting for testing an advanced FSM framework.

To address the identified gaps, this study introduces and validates a novel integrated framework built on three core innovations. First, we apply a strict frequency-based consensus rule across nine different feature selection methods to identify a robust and definitive set of predictor variables from GEE data. Second, we carry out a systematic comparative optimization of an LSTM network using five metaheuristic algorithms, including GWO, WOA, Crow Search Algorithm (CSA), OOA, and HOA, to benchmark their performance for FSM. Third, we bring these two stages together into a single, cohesive workflow, where the consensus-derived feature set directly informs the hyperparameter optimization process. The ultimate goal is to develop a highly accurate, reliable, and practical FSM tool, particularly suited for arid and semi-arid regions facing growing hydrological extremes.

## 2. Materials and methods

### 2.1. Study area

Khuzestan Province in southwestern Iran, forming part of the low-lying Mesopotamian Plain, was selected as the study area due to its well-documented history of severe and recurrent flooding [20]. Its flat topography, drained by major rivers (Karun and Karkheh), creates a natural floodplain susceptible to inundation. The climate paradoxically heightens flood risk, as infrequent but intense rainfall events, combined with water infrastructure challenges, lead to catastrophic floods. Major events in 2015, 2019, and 2022 caused significant socio-economic and environmental damage, devastating agriculture, and damaging critical infrastructure [21]. As Iran's primary agricultural and hydrocarbon hub, and home to ecologically vital areas such as the Shadegan Wetland (Ramsar site), Khuzestan's combination of high exposure, economic importance, and ecological sensitivity makes it an ideal testbed for developing an advanced FSM framework (Fig 1).

### 2.2. Data Preparation

**2.2.1. Environmental Factors.** Nineteen initial environmental factors were derived from global datasets, primarily accessed and processed via the GEE platform for the period 2000–2023. Climatic variables were sourced from the TerraClimate dataset, soil properties from SoilGrids, and topographic indices from the SRTM digital elevation model. The NDVI was calculated from MODIS imagery. All raster layers were matched to a common coordinate system (WGS84), resampled to a 1-km spatial resolution, and masked to the study area boundary. A statistical summary of these prepared factors is presented in Table 1.

**2.2.2. Flood inventory and data partitioning.** A binary flood inventory map was created by combining historical flood event polygons from the Khuzestan Provincial Disaster Management Organization and water extent maps derived from Sentinel-1 SAR imagery for major flood years (2015, 2019, 2022). Non-flood locations were randomly sampled from areas outside these inundation zones, yielding a total of 27,515 sample points.

For model development, a balanced subset of 1,000 sample points (500 flood, 500 non-flood) was used to ensure class balance and stable training. This subset was randomly partitioned into 70% for training (700 points) and 30% for independent testing (300 points). The confusion matrices and evaluation metrics reported in Section 3.3 are based on these 300 test samples. After confirming model stability and accuracy, the trained model was subsequently generalized to the full dataset of 27,515 points to produce the final flood susceptibility map.

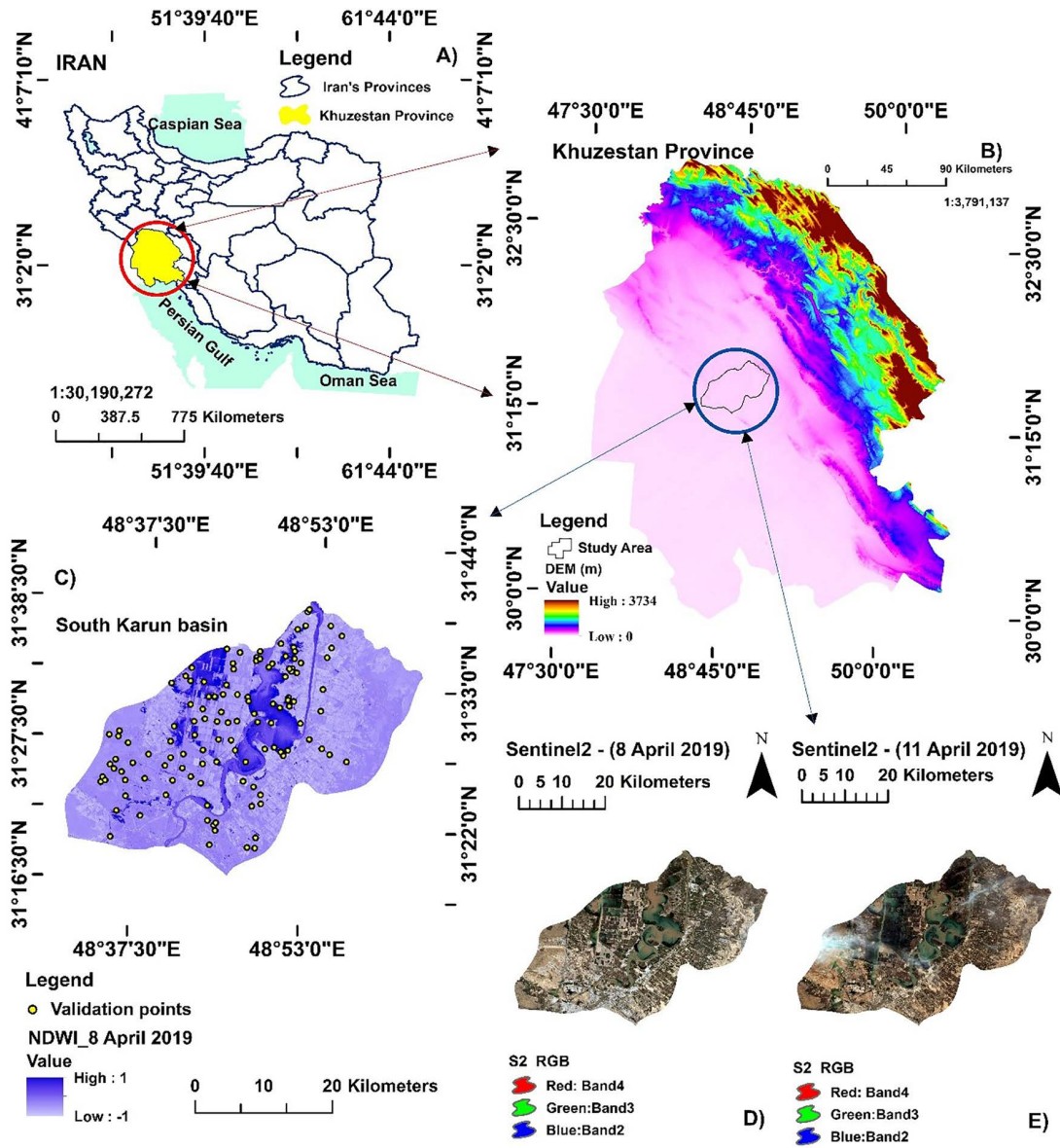

**Fig 1. The location of the study area in Iran and in Khuzestan province, generated by the authors using Sentinel-2 satellite data (Copernicus Open Access License) processed in Google Earth Engine (https://cloud.google.com) and QGIS software (https://qgis.org).**

## 2.3. Ensemble feature selection

To identify a robust and non-redundant set of predictors, an ensemble of nine diverse feature selection methods was applied to the training data (Table 2). This approach mitigates the bias of any single method. A majority-vote consensus rule was implemented. A variable was retained in the final optimal set only if it was selected by at least five of the nine methods. This consensus-based feature set was used for all subsequent modeling.

**Table 1. Statistical properties of the environmental factors.**

| Variable Name | Unit | Abbreviation | Minimum | Maximum | Mean | Standard Deviation |
|---|---|---|---|---|---|---|
| Actual Evapotranspiration | mm | Aet | 13.00 | 22.20 | 17.23 | 2.43 |
| Slope Aspect | – | Aspect | 1.00 | 10.00 | 5.00 | 2.74 |
| Soil Bulk Density | kg/m³ | Bulk | 0.00 | 169.00 | 147.89 | 22.16 |
| Soil Clay Content | % | Clay | 0.00 | 37.00 | 23.52 | 5.21 |
| Soil Water Deficit | mm | Def | 127.80 | 158.10 | 144.62 | 7.68 |
| Elevation | m | DEM | 0.00 | 106.00 | 23.28 | 4.62 |
| Vegetation Index | – | NDVI | −0.45 | 0.69 | 0.09 | 0.168 |
| Land Cover | – | Land Cover | 1.00 | 13.00 | 8.00 | 3.92 |
| Palmer Drought Index | – | PDSI | 1.60 | 2.70 | 1.93 | 0.25 |
| Reference Evapotranspiration | mm | Pet | 149.60 | 171.60 | 161.80 | 5.83 |
| Cumulative Precipitation | mm | Pr | 12.00 | 21.00 | 15.81 | 2.54 |
| Soil Sand Content | % | Sand | 0.00 | 72.00 | 43.56 | 9.63 |
| Soil Texture | – | Soil texture | 0.00 | 9.00 | 6.00 | 1.70 |
| Surface Soil Moisture | mm | Soil moisture | 5.00 | 13.19 | 9.01 | 1.94 |
| Daily Minimum Temperature | °C | TMMN | 15.50 | 17.29 | 16.56 | 0.47 |
| Daily Maximum Temperature | °C | TMMX | 30.40 | 32.50 | 31.56 | 0.50 |
| Evapotranspiration | kg/m²/s | FLDAS | 0.22 | 0.40 | 0.29 | 0.07 |
| Slope | % | Slope | 0 | 24.61 | 1 | 1.04 |
| Landscape | – | Landscape | 1 | 13 | 7 | 4 |

**Table 2. Ensemble feature selection methods.**

| Method | Category | Mechanism and Principle |
|---|---|---|
| Boruta | Wrapper | Uses Random Forest and statistical testing against random "shadow" features to identify crucial variables [22]. |
| Boruta-SHAP | Wrapper | Extends Boruta by using SHapley Additive exPlanations (SHAP) values for a model-agnostic importance score. |
| Elastic-Net | Embedded | A regularization method combining L1 (Lasso) and L2 (Ridge) penalties for automatic feature selection [23]. |
| Mutual Information (MI) | Filter | Estimates the non-linear statistical dependence between each feature and the target variable. |
| Permutation Importance | Model-Agnostic | Measures the increase in a model's prediction error after randomly permuting a feature's values. |
| Recursive Feature Elimination (RFE) | Wrapper | Recursively removes the least important features based on a model's performance. |
| Sequential Forward Selection (SFS) | Wrapper | Greedily adds features one by one to maximize a chosen performance metric (e.g., R²). |
| Stability Selection | Hybrid | Combines subsampling with a base selector to identify features stable across data perturbations [24]. |
| Deep Feature Importance | Embedded | Derives feature importance from the weights of a simple Artificial Neural Network (ANN). |

## 2.4. LSTM model development and hyperparameter optimization

**2.4.1. LSTM.** An LSTM (Table 3) was implemented using the Keras API in TensorFlow [6]. The model's performance is susceptible to several architectural hyperparameters, which constituted the search space for

**Table 3. LSTM Cell: Gate Equations and State Update Mechanisms.**

| Gate | Equation | Description |
|------|----------|-------------|
| Forget Gate | $f_t = \sigma(W_f \cdot [h_{t-1}, x_t] + b_f)$ | Decides what to discard from the cell state |
| Input Gate | $i_t = \sigma(W_i \cdot [h_{t-1}, x_t] + b_i)$ | Decides which new information to store |
| Candidate State | $\widetilde{C}_t = \tanh(W_C \cdot [h_{t-1}, x_t] + b_C)$ | Candidate values to add to the cell state |
| Cell State Update | $C_t = f_t * C_{t-1} + i_t * \widetilde{C}_t$ | Updates the long-term memory |
| Output Gate | $o_t = \sigma(W_o \cdot [h_{t-1}, x_t] + b_o)$ | Decides what to output from the cell state |
| Hidden State | $h_t = o_t * \tanh(C_t)$ | Outputs the short-term memory |

Where $\sigma$ is the sigmoid function, $*$ denotes element-wise multiplication, and $W$ and $b$ are weight matrices and bias vectors, respectively.

optimization, including Number of LSTM units per layer: [10, 200], Number of LSTM layers: [1], and Learning Rate: [0.001, 0.05] (log scale).

The LSTM model was implemented with the following architecture: an input layer (sequenceInputLayer(numFeatures)), an LSTM layer with 100 hidden units and sequence output mode (lstmLayer(100, 'OutputMode','sequence')), a fully connected layer (fullyConnectedLayer(1)), and a regression layer with mean squared error (MSE) loss. Training was conducted using the Adam optimizer with a maximum of 100 epochs, an initial learning rate of 0.01 with a piecewise schedule (drop factor of 0.2 every 125 epochs), a gradient threshold of 1, and a batch size of 128. No early stopping or validation set was applied, and data shuffling was performed every epoch.

**2.4.2 Metaheuristic Optimization Process and Configuration.** Five metaheuristic algorithms were employed to navigate the hyperparameter search space (Table 4). Each algorithm uses a unique biologically-inspired strategy to refine hyperparameter combinations. All optimizers were configured with a population size of 10 and executed for 20 iterations.

**2.4.3. Spatial application of LSTM.** LSTM is employed for spatial FSM rather than temporal sequence forecasting. Each pixel is treated as an independent sample, and all environmental predictors, including static spatial features (e.g., elevation, slope, soil properties) and long-term aggregated climatic indices (e.g., mean NDVI, cumulative Pr), are compiled into a single static feature vector per pixel. This vector is input into the LSTM network, which uses its gating mechanisms to learn complex nonlinear interactions among spatial variables and assign optimal weights to each feature. The model outputs a susceptibility probability for each pixel. Thus, the LSTM's sequential memory is repurposed to capture contextual dependencies within the feature space rather than temporal dependencies across time steps. This spatial application of LSTM is validated by the high predictive performance achieved and is now clearly described to ensure transparency and reproducibility.

## 2.5. Model evaluation, validation, and stability assessment

A comprehensive suite of evaluation metrics was employed to rigorously assess model performance across multiple dimensions, including classification accuracy, cost sensitivity, regression quality, and statistical significance. This validation framework ensures robust model comparison and selection for FSM applications (Table 5).

To assess the statistical stability and reproducibility of the proposed LSTM-WOA model, a repeated k-fold cross-validation was conducted. Specifically, 5-fold cross-validation was repeated 15 times (total 75 independent runs). For each fold, the model was trained on 80% of the data and evaluated on the remaining 20%, with stratification to preserve class balance. The mean, standard deviation, coefficient of variation (CV), and 95% confidence intervals were calculated for key performance metrics (Accuracy, AUC, F1-Score, Precision, Recall, and Kappa) to evaluate model consistency across different data partitions.

**Table 4. Metaheuristic Algorithms for LSTM Hyperparameter Optimization.**

| Algorithm (Abbrev.) | Formula | Key Parameters | Description |
|---|---|---|---|
| **Grey Wolf Optimizer (GWO)** | $\vec{X}(t+1) = \frac{1}{3}(\vec{X}_1 + \vec{X}_2 + \vec{X}_3)$ where $\vec{X}_1 = \vec{X}_\alpha - \vec{A}_1 \cdot \| \vec{C}_1 \vec{X}_\alpha - \vec{X} \|$ | • Population: 10<br>• Max Iterations: 20<br>• Convergence (a): Linear 2→0 | Follows the three best agents (α, β, δ wolves) in a pack hierarchy [25]. |
| **Whale Optimization Algorithm (WOA)** | $\vec{X}(t+1) = \begin{cases} \vec{X}^* - \vec{A} \cdot \| \vec{C}\vec{X}^* - \vec{X} \| & \text{if } p < 0.5 \\ \| \vec{X}^* - \vec{X} \| \cdot e^{bl} \cdot \cos(2\pi l) + \vec{X}^* & \text{otherwise} \end{cases}$ | • Population: 10<br>• Max Iterations: 20<br>• a: 2→0, b: 1, p: 0.5 | Simulates whale bubble-net hunting via encircling or spiral movement [26]. |
| **Crow Search Algorithm (CSA)** | $\vec{X}_i^{t+1} = \vec{X}_i^t + r_i \cdot fl \cdot (\vec{m}_j^t - \vec{X}_i^t)$, if $r_j \geq AP$ | • Population: 10<br>• Max Iterations: 20<br>• fl: 2.0, AP: 0.1 | Solutions probabilistically follow others memorized best positions [27]. |
| **Orangutan Optimization Algorithm (OOA)** | $\vec{X}_i^{t+1} = \vec{X}_i^t + \alpha \cdot (\vec{X}_{best} - \vec{X}_i^t) + \beta \cdot (\vec{X}_{rand} - \vec{X}_i^t)$ | • Population: 10<br>• Max Iterations: 20<br>• α: 0.7, β: 1.5 | Combines movement toward global best with random foraging exploration [18]. |
| **Hippopotamus Optimization Algorithm (HOA)** | $\vec{X}_i^{t+1} = \vec{X}_{center} + D \cdot \vec{r}$ | • Population: 10<br>• Max Iterations: 20<br>• D: 1.8 | Focuses on intensive local search around promising defense zones [19]. |

**Table 5. Model Evaluation Metrics with Parameter Descriptions.**

| Metric Category | Metric Name and Formula | Parameters | Purpose |
|---|---|---|---|
| ROC Analysis | Area Under Curve (AUC)<br>Derived from the ROC plot | TPR: True Positive Rate<br>FPR: False Positive Rate | Measures overall classification power. Higher AUC (0.5–1.0) indicates better discrimination between flood and non-flood areas [28]. |
| Binary Classification Rates | True Positive Rate (TPR)/ Recall<br>$TPR = \frac{TP}{TP+FN}$ False Positive Rate (FPR)<br>$FPR = \frac{FP}{FP+TN}$ | TP: True Positives<br>FN: False Negatives<br>FP: False Positives<br>TN: True Negatives | TPR: Model's sensitivity to detect actual floods. FPR: Rate of false alarms where non-flood areas are incorrectly classified. |
| Cost-Sensitive Analysis | Normalized Expected Cost – NE(C)<br>$NE(C) = \frac{(1 - TP \cdot P(+) \cdot C(-\|+) + FP \cdot P(-) \cdot C(+\|-))}{P(+) \cdot C(-\|+) + P(-) \cdot C(+\|-)}$ | P(+): Prior probability of flood<br>P(-): Prior probability of non-flood<br>C(-\|+): Cost of missing a flood<br>C(+\|-): Cost of false alarm | Evaluates practical cost-effectiveness under class imbalance. Lower NE(C) indicates better economic efficiency in decision-making [29]. |
| Classification Metrics | Precision<br>$Precision = \frac{TP}{TP+FP}$ F1-Score<br>$F1 = 2 \times \frac{Precision \times Recall}{Precision+Recall}$ Cohen's Kappa (κ)<br>$\kappa = \frac{P_o - P_e}{1 - P_e}$ | $P_o$: Observed agreement proportion<br>$P_e$: Expected chance agreement | Precision: Reliability of flood predictions. F1: Balanced measure of precision and recall. κ: Agreement beyond random chance (0–1 scale) [30]. |
| Regression Metrics | Root Mean Squared Error (RMSE)<br>$RMSE = \sqrt{\frac{1}{n} \times \sum_{i=1}^{n}\left[(x_{imeans} - x_{ipred})^2\right]}$<br>Mean Absolute Error (MAE)<br>$MAE = \frac{1}{n} \times \sum_{i=1}^{n}|x_{imeas} - x_{ipred}|$ | n: Sample count<br>$x_{meas}$: Measured value<br>$x_{pre9}$: Predicted value | RMSE: Penalizes large prediction errors. MAE: Average absolute error magnitude. Lower values indicate better accuracy. |
| Correlation Metrics | Coefficient of Determination (R²)<br>$R^2 = 1 - \frac{\sum_{i=1}^{n}(x_{meas} - x_{pred})^2}{\sum_{i=1}^{n}(x_{meas} - \bar{x})^2}$<br>Correlation Coefficient (R)<br>$R = \frac{\sum (x_{meas} - \bar{x})(x_{pred} - \bar{x}_{pred})}{\sqrt{\sum (x_{meas} - \bar{x})^2 \cdot \sum (x_{pred} - \bar{x}_{pred})^2}}$ | $\bar{x}_{meas}$: Mean of measured values<br>$\bar{x}_{pre9}$: Mean of predicted values | R²: Proportion of variance explained (0–1). R: Linear correlation strength (−1–1). Higher values indicate better model fit. |
| Statistical Test | Friedman Test Statistic<br>$F_r = \frac{12}{nk(k+1)} \sum_{i=1}^{k} R_i^2 - 3(k+1)n$ | n: Number of datasets/folds<br>k: Number of models<br>$R_i$: Sum of ranks for model i | Non-parametric test for comparing multiple models. Determines if performance differences are statistically significant [31,32]. |

## 2.6. Flood susceptibility mapping

The final FSM was generated by applying the best-performing model, LSTM-WOA, using the consensus-derived feature set (NDVI, TMMN, Pr, Soil moisture, Aet, Def, TMMX, FLDAS, PDSI, DEM, Pet, Soil texture, Sand, and Clay) to compute a continuous susceptibility index across the study area, which was then classified into five susceptibility classes (Very Low to Very High).

The four-stage workflow of the present study is included: 1) environmental data and flood inventory preparation, 2) consensus-based feature selection from nine methods, 3) LSTM development and hyperparameter optimization with five metaheuristic algorithms (WOA, OOA, GWO, CSA, HOA), and 4) comprehensive model validation leading to the selection of LSTM-WOA for final mapping (Fig 2).

# 3. Results

## 3.1. The feature selection results

After obtaining the relevant variables from GEE, feature selection was done to determine the most important variables influencing the modeling process. Using the Boruta method, only 10 out of the initial 19 variables were selected as the chosen features. The most important variables, in order, are including Pr, TMMN, NDVI, PDSI, Def, Soil moisture, TMMX, FLDAS, Aet, and DEM (Fig 3). Hydrological dynamics are controlled by climatic and soil-vegetation moisture variables (Pr, temperature, NDVI, PDSI, Soil moisture, Aet), not static topography. Flat terrain favors vertical processes (infiltration, FLDAS) over surface routing. Pr is the key input; high TMMN/TMMX ranks reflect ET dominance. NDVI and PDSI capture vegetation and drought effects on Soil moisture. Moisture-related variables (Soil moisture, Def, Aet, FLDAS) link atmospheric forcing to flood generation via infiltration, storage, and FLDAS.

In the Boruta-SHAP method, 10 out of the total 19 variables were selected as the most influential variables for the modeling process. According to this method, the selected variables, in order, are NDVI, Pr, DEM, Soil moisture, Aet, TMMN, Def, FLDAS, PDSI, and TMMX (Fig 4). The nature of the selected variables remains unchanged compared to the Boruta method; however, the priority of influence of these variables differs in this method compared to the standard Boruta. Compared to Boruta, Boruta-SHAP provides a process-based hierarchy showing that vegetation mediates how climatic inputs translate into flood response, which is key for nonlinear modeling.

The Elastic-Net method selected a total of 9 variables as influential features. These variables, in order of importance, include NDVI, TMMN, Aet, TMMX, Pet, Pr, and Soil moisture (Fig 5). The DEM variable occupies an intermediate position and is not strongly selected. Unlike the previous two methods, Elastic-Net emphasizes the hydrological balance between atmospheric demand, actual water use, and storage capacity. The inclusion of Pet alongside Aet and Pr quantifies environmental water stress. Soil texture emerges as a key control suggests those techniques failed to capture subtle linear effects. DEM remains unimportant, confirming climatic dominance over topography in this flat system. The selected variables depict a coherent semi-closed hydrological system where water is partitioned among atmospheric demand (Pet), vegetation use (Aet, NDVI), input (Pr), and substrate storage (Soil texture, Soil moisture). This linear, process-oriented view complements the nonlinear insights from Boruta-SHAP.

Based on the Mutual Information method, the selected variables, Sequentially, are NDVI, Def, Pet, Soil moisture, Aet, TMMN, FLDAS, TMMX, Pr, and PDSI (Fig 6). Unlike the Elastic-Net, this algorithm selected Def, while the PDSI variable was borderline and selected intermediately. In this method, 10 variables were selected for input into the learning processes. Mutual Information ranks variables by their total dependence on flood response. NDVI, Def, Pet, Soil moisture, and Aet occupy the highest positions. This indicates that the region's hydrology is governed by the soil–vegetation system and evapotranspiration dynamics. These five variables represent atmospheric demand, actual water loss, available Soil moisture, and moisture deficit. Their dominance reflects the rapid and nonlinear behavior of the unsaturated zone. Pr ranks lower because its effect on flooding is mediated through Soil moisture and vegetation rather than a direct link. DEM

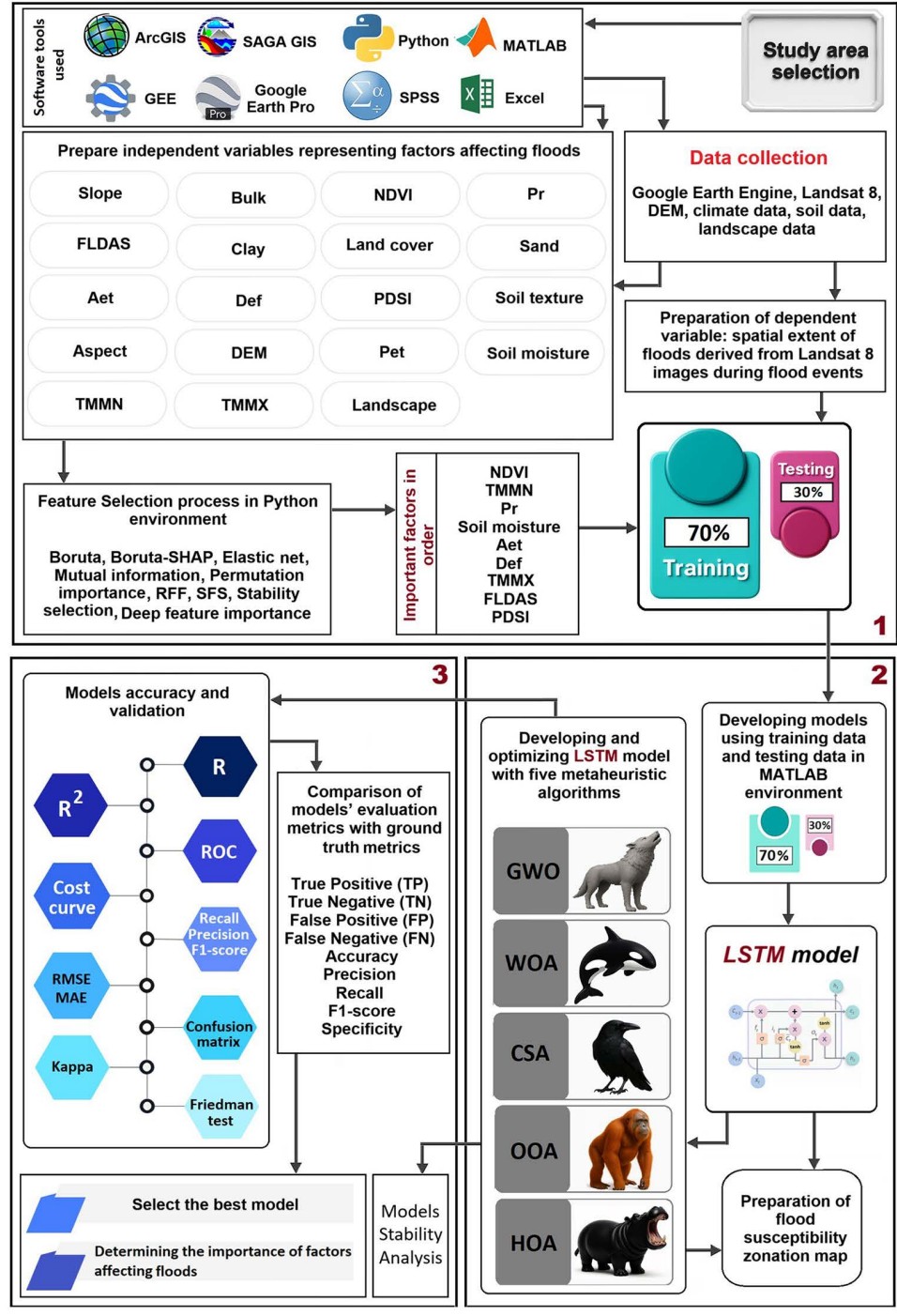

**Fig 2. A Schematic Diagram of the Research Methodology.** Original figure created by the authors using Adobe Photoshop (https://www.adobe.com).

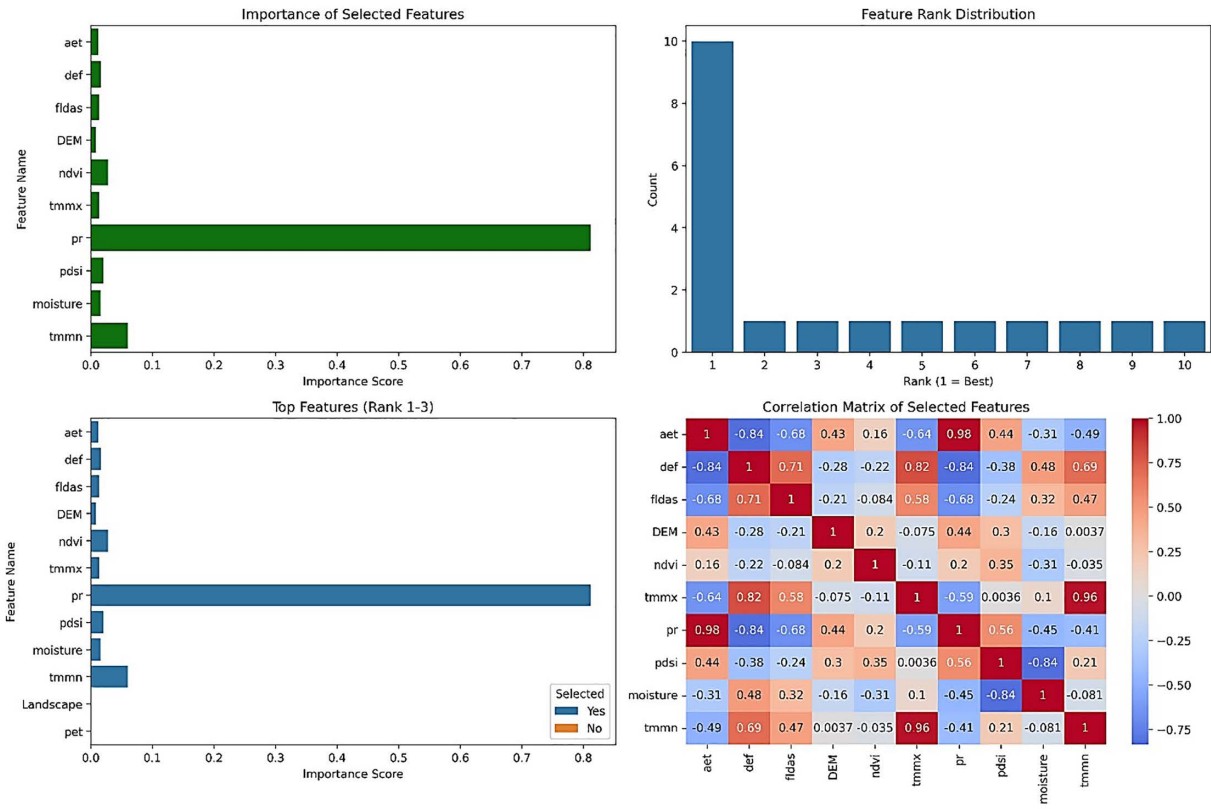

**Fig. 3.** Boruta feature selection results, generated in google colab (https://colab.research.google.com).

remains negligible, confirming topographic control is minimal in this flat setting. The results emphasize that the balance between water supply, atmospheric demand, and plant uptake defines the hydrological regime. This ranking provides critical guidance for selecting monitoring variables and predictors in machine learning models.

Based on the Permutation Importance method, the selected variables are NDVI, Def, Pr, Sand, Clay, DEM, TMMN, Aet, Soil moisture, and TMMX (Fig 7). Sand and Clay were selected as important for the modeling processes for the first time, as they had not been chosen by any previous algorithms. Permutation Importance measures the drop in model accuracy when a variable is shuffled. NDVI ranks highest, confirming its role in linking water, soil, and vegetation. Sand and Clay capture nonlinear effects on infiltration and water retention in flat terrain. Def ranks above Pr, meaning moisture deficit predicts flood response better than rainfall alone. Temperature variables highlight evapotranspiration control. DEM captures micro-topographic effects on ponding and soil moisture. These results show that modeling Soil texture, Soil moisture, and vegetation together is essential for understanding nonlinear flood responses in low-relief areas.

Based on the RFE method, the selected variables are Pr, TMMN, NDVI, PDSI, Def, Soil moisture, FLDAS, TMMX, and Aet (Fig 8). Among these, the roles of Pr, TMMN, and NDVI are emphasized as more significant than those of the other variables. In this method, 9 variables were selected as influential features. Ref confirms Pr as the main flood driver, followed by minimum temperature (TMMN), which controls antecedent Soil moisture through lower nighttime evapotranspiration. NDVI ranks third, reflecting vegetation effects like interception and infiltration that slow runoff. PDSI and Def represent pre-storm wetness and soil storage, determining whether rainfall infiltrates or becomes runoff. Soil moisture, Aet, and TMMX capture near-event water status and evaporative loss. DEM ranks low, consistent with flat terrain where

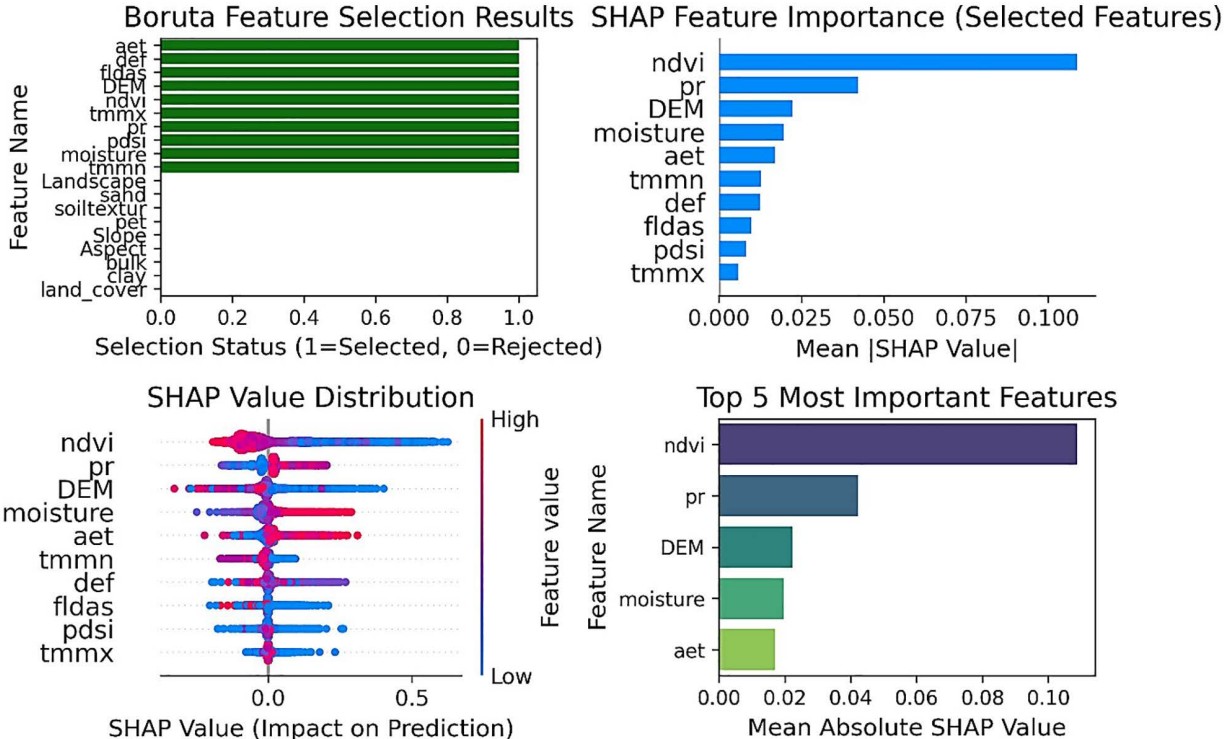

**Fig 4. Boruta-SHAP Value Distribution Feature Selection Results, generated in Google Colab (https://colab.research.google.com).**

vertical processes dominate. Together, these variables form a cascading system: Pr is modulated by moisture deficits, vegetation, and temperature, which all influence soil storage and flood response.

The SFS method showed the maximum coefficient of determination by selecting 7 variables as the most influential, in the following order: NDVI, TMMN, Soil texture, Soil moisture, Sand, Aspect, and Clay (Fig 9). This method operated significantly differently compared to previous approaches, because Soil texture and Aspect were selected as important for the first time. Additionally, this method reaffirmed the importance of NDVI and TMMN. SFS identifies the physical foundation of flood response: soil texture controls infiltration and storage, Aspect captures microclimatic energy, NDVI integrates soil-vegetation-moisture, and TMMN reflects nighttime ET and moisture retention. Pr and FLDAS are absent because their effects are already embedded in vegetation and soil moisture. Stability Selection further distills this to three core controls (NDVI, TMMN, and TMMX), showing that flood generation in this plain is governed by vegetation and daily temperature range, which integrate the cumulative effects of rainfall, drought, and evaporative demand. Pr and Soil moisture still matter, but their influence is fully encoded in these stable variables. Thus, vegetation and temperature alone offer a simple yet physically meaningful basis for flood modeling and forecasting in this region.

In the Stability Selection method, only three variables were selected as important and influential, while the rest were not chosen. As shown in Fig 10, the selected variables are NDVI, TMMN, and TMMX. Deep learning identifies NDVI and Aet as dominant controls, confirming vegetation-driven evapotranspiration as the primary water output. DEM gains relevance, capturing micro-topographic effects on ponding and moisture distribution. Pr ranks low; its influence is indirect, mediated through soil-vegetation states.

A stable core of hydrological controls persists across all techniques: NDVI, Pr, and temperature (TMMN, TMMX). NDVI is universally central, integrating cumulative water balance. Pr drives the system, but its signal is transmitted indirectly

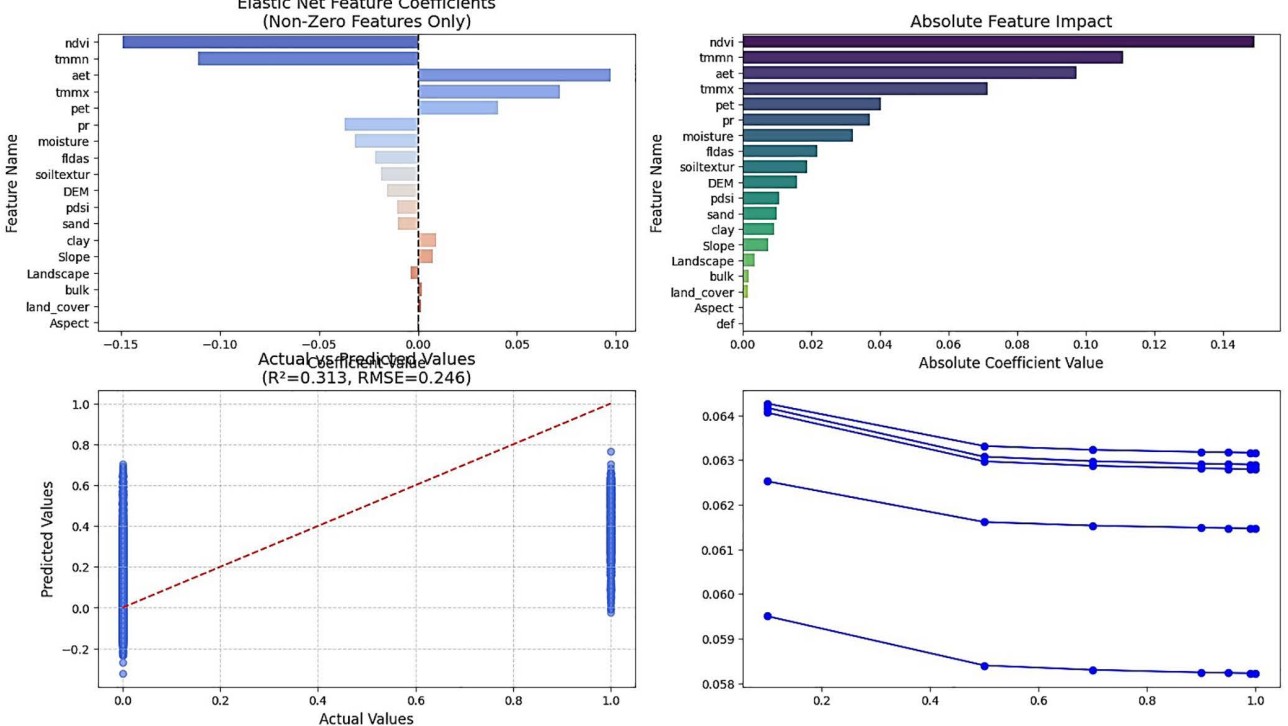

**Fig 5. Selection of Important Variables Using the Elastic-Net Method.** Original figure created by the authors (output from analysis in Google Colab (https://colab.research.google.com)).

via soil moisture and vegetation. A hierarchical modeling approach is recommended: (1) core stable variables (NDVI, Pr, TMMN, TMMX); (2) intermediate moisture variables (Soil moisture, Def, Aet); (3) intrinsic landscape properties (Soil texture, Aspect, DEM). This framework couples climatic forcing, ecohydrological mediation, and a physical template for robust flood mapping in flatlands.

The ANN method is used for the important variables in the learning process, illustrating the learning stages and the key variables. In this method, the variables are ranked in the following order of importance, including NDVI, Aet, DEM, TMMN, Soil moisture, FLDAS, and, to a lesser extent, Pr (Fig 11). The ANN ranks NDVI and Aet highest, confirming vegetation-driven evapotranspiration as the dominant water loss; lower Pr importance reflects its indirect, soil-vegetation mediated effect on flood response.

Different feature selection methods have introduced various variables for input into the modeling process, often with significant discrepancies. To derive the most reliable outcome, the intersection of results from all models was utilized. Table 6 provides a comparative overview of the variables selected by each feature selection method.

The analysis of variable selection frequency across nine distinct methods revealed a clear hierarchy of predictor importance (Table 7). Two variables, NDVI and TMMN, were selected by all nine methods, establishing them as the most robust and critical features in the model. A second tier of variables, including Pr and Soil moisture (selected 7/9 times), along with Aet, Def, TMMX, and FLDAS (selected 6/9 times), were strongly to moderately selected, confirming their consistent relevance. In contrast, a large group of variables, such as PDSI, DEM, Pet, and soil properties (Soil texture, Sand, Clay), were only weakly selected (3–5/9 times). Finally, topographic features (Slope, Aspect), Bulk, Land Cover, and Landscape were effectively eliminated from consideration, being selected once or not at all, indicating they contribute negligible explanatory power to the models in this analysis.

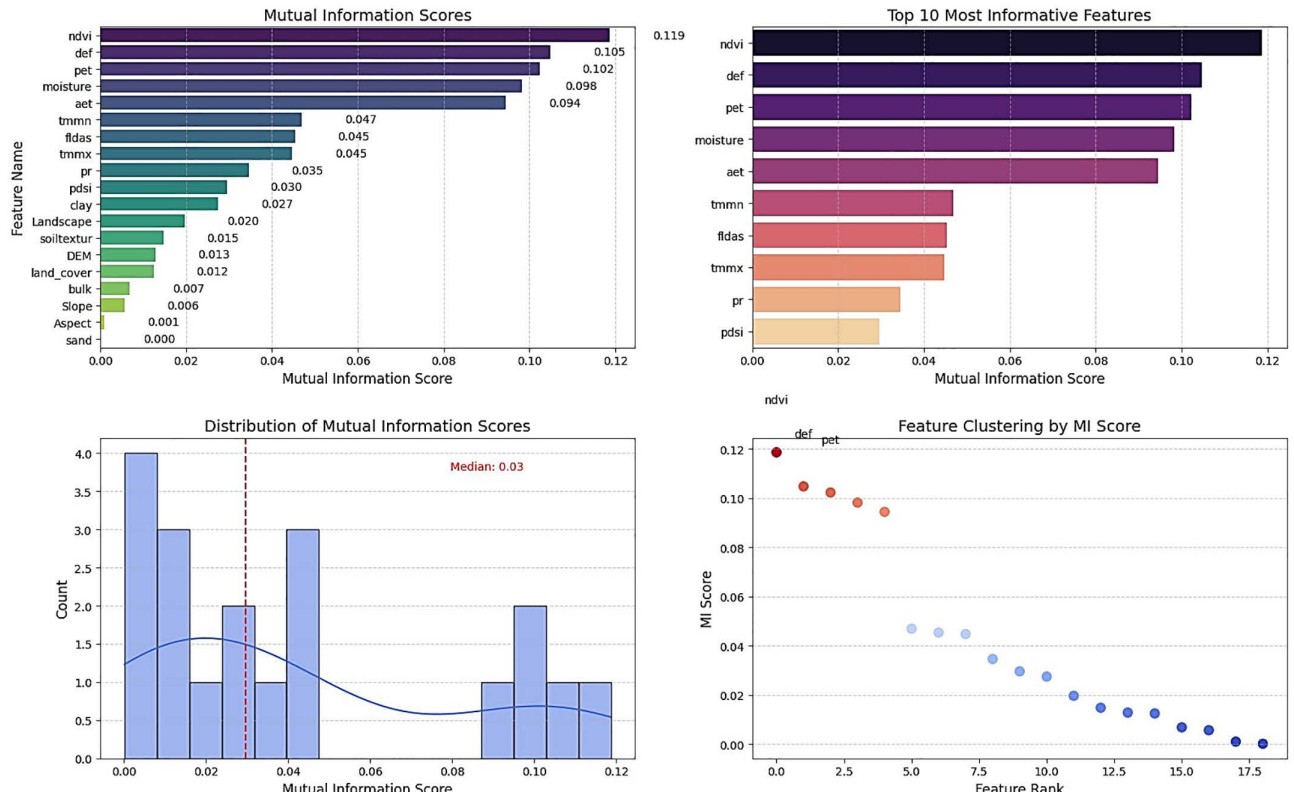

**Fig 6. Selection of Important Variables Using the Mutual Information Method, generated in Google Colab (https://colab.research.google.com).**

## 3.2. Modeling with DLMs

The resulting FSMs for the study area indicate that the highest susceptibility to flooding is concentrated in the northwestern and central parts of the region. These areas were consistently identified across the optimized models as having the highest potential for flood occurrence (Fig 12). This spatial pattern reflects topographic controls and downstream flow accumulation, where even subtle low-gradient conditions in flat terrain promote prolonged inundation and surface water convergence.

## 3.3. Evaluation of modeling accuracy and stability

The results from the confusion matrices indicate that the base LSTM model performs adequately, with an accuracy of 92.6% for the negative class and 80.9% for the positive class, while maintaining a relatively low error rate in predicting positive samples. The LSTM-GWO model reduced false positives compared to the base model but introduced more errors in the negative class. In contrast, the LSTM-WOA model achieved a suitable balance between both classes, demonstrating balanced and stable performance with 90.5% accuracy for the negative class and 91.4% for the positive class.

On the other hand, the LSTM-CSA model recorded the highest accuracy in the negative class (94.6%), but its accuracy in the positive class decreased to 71.1%, indicating a tendency toward predicting the negative class and reduced sensitivity. The LSTM-OOA model achieved the highest accuracy in the positive class (93.4%), but its accuracy in the negative class was 75.7%, reflecting reduced specificity. The LSTM-HOA model showed performance similar to the base model, with 92.6% accuracy for the negative class and 79.6% for the positive class, albeit with slight improvement in predicting positive instances.

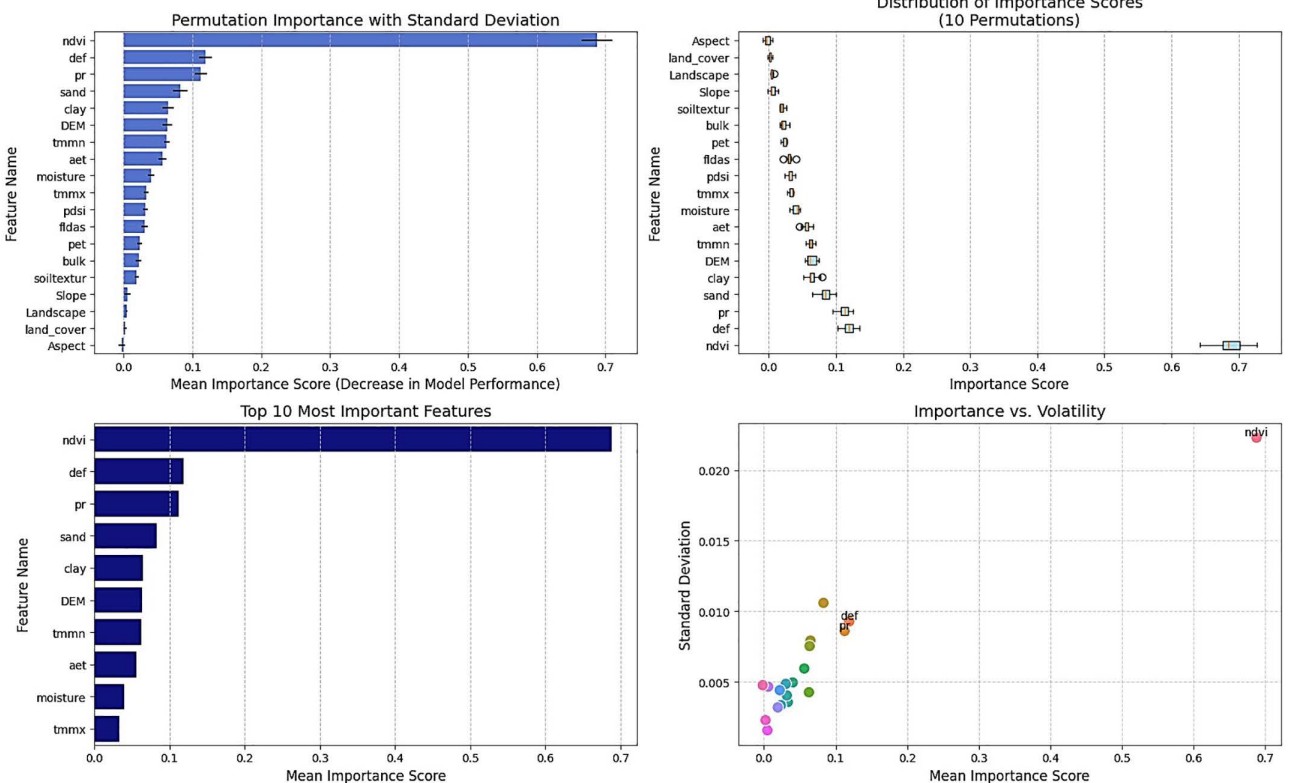

**Fig 7. Selection of Important Variables Using the Permutation Importance Method, generated in Google Colab (https://colab.research.google.com).**

Considering the balance between accuracy in both classes and the avoidance of bias toward either class, the LSTM-WOA model delivered the best performance among all evaluated models. Figure 13 displays the confusion matrices for the base LSTM model and its five metaheuristic-optimized versions. These matrices present the prediction accuracy for both positive and negative classes, along with percentage values. Based on them, performance metrics such as overall accuracy, recall (sensitivity), and specificity can be compared (Fig 13). The balanced performance of LSTM-WOA reflects improved hydrological discrimination, capturing the nonlinear transition from non-flooded to flooded states without overfitting to either class, which is critical for reliable flood hazard mapping under varying antecedent moisture conditions.

To comprehensively evaluate the performance of the LSTM model and its optimized versions in predicting FSMs, the Area Under the Cost Curve (AUCC) metric was utilized (Fig 14). The base LSTM model achieved an AUCC of 0.10, reflecting acceptable yet suboptimal performance in minimizing prediction costs. The application of metaheuristic algorithms significantly improved this metric. Among the optimized models, LSTM-WOA and LSTM-HOA recorded the best performance, each with an AUCC of 0.08, demonstrating their superior ability to reduce classification costs under uncertain conditions. The LSTM-GWO, LSTM-CSA, and LSTM-OOA models also showed improvement compared to the base model, with an AUCC of 0.09, although their enhancement was less pronounced compared to WOA and HOA. The superior AUCC of LSTM-WOA and LSTM-HOA indicates better simulation of the hydrological memory effect, where antecedent moisture and cumulative rainfall control flood probability, thereby reducing costly misclassifications in transitional saturated zones.

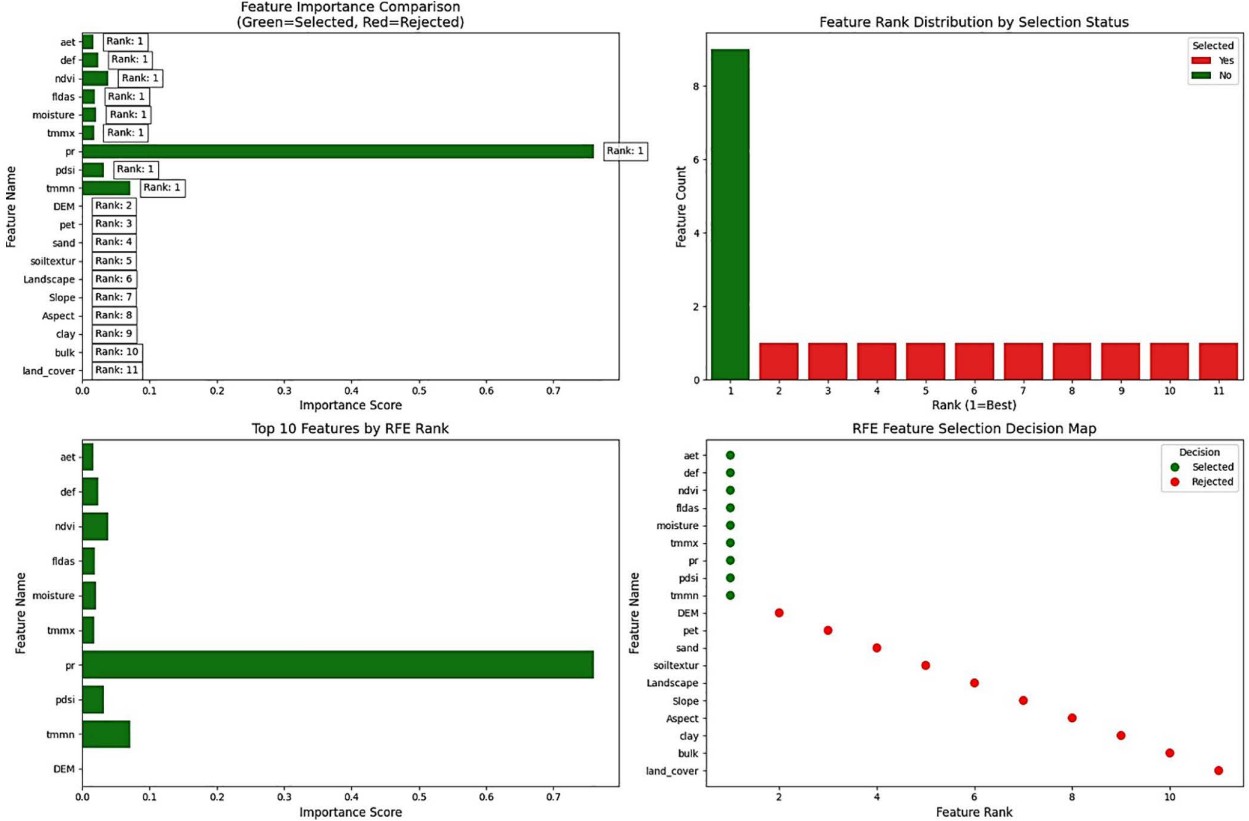

**Fig 8. Selection of Important Variables Using the RFE Method, generated in Google Colab (https://colab.research.google.com).**

Metaheuristic optimization has shifted the cost curve toward lower values and reduced the area beneath it (Fig 15). This indicates that the optimized models make more accurate and efficient decisions across various cost and threshold conditions. Optimized LSTM models (GWO-LSTM, WOA-LSTM) outperform the baseline LSTM, achieving lower normalized cost values (0.08–0.10). This improved accuracy reflects better simulation of nonlinear hydrological processes in this flatland system—particularly time lags between rainfall and runoff, gradual soil moisture accumulation, and vegetation-mediated responses. These models more effectively capture long-term dependencies and system memory, such as antecedent wetness and drought legacy effects. The reduction in prediction error, though modest, translates to improved identification of local flood-prone areas, supporting more reliable hazard mapping and early warning. Hybrid deep learning frameworks thus offer enhanced generalization for flood prediction under complex hydroclimatic conditions.

Based on the results presented in Table 8, the performance of the base LSTM model and its optimized versions using five metaheuristic algorithms was compared for FSMs. In terms of Precision and Recall, the LSTM-OOA model achieved the highest Recall value (0.93), while the LSTM-CSA model demonstrated the best Precision (0.93). Regarding the combined metric F1-Score, which balances Precision and Recall, the LSTM-WOA model outperformed all others with a value of 0.8825. Additionally, the highest Cohen's Kappa value was recorded for LSTM-WOA (0.75), indicating that its predictions align with actual results significantly beyond random chance. In the continuous statistical metrics section, the LSTM-WOA model achieved the highest correlation coefficient R (0.78), and its $R^2$ value (0.57) was relatively high, reflecting its superior ability to explain data variability. In terms of error metrics, the LSTM-GWO model exhibited the lowest MAE (0.22) and RMSE (0.32) values, indicating higher prediction accuracy.

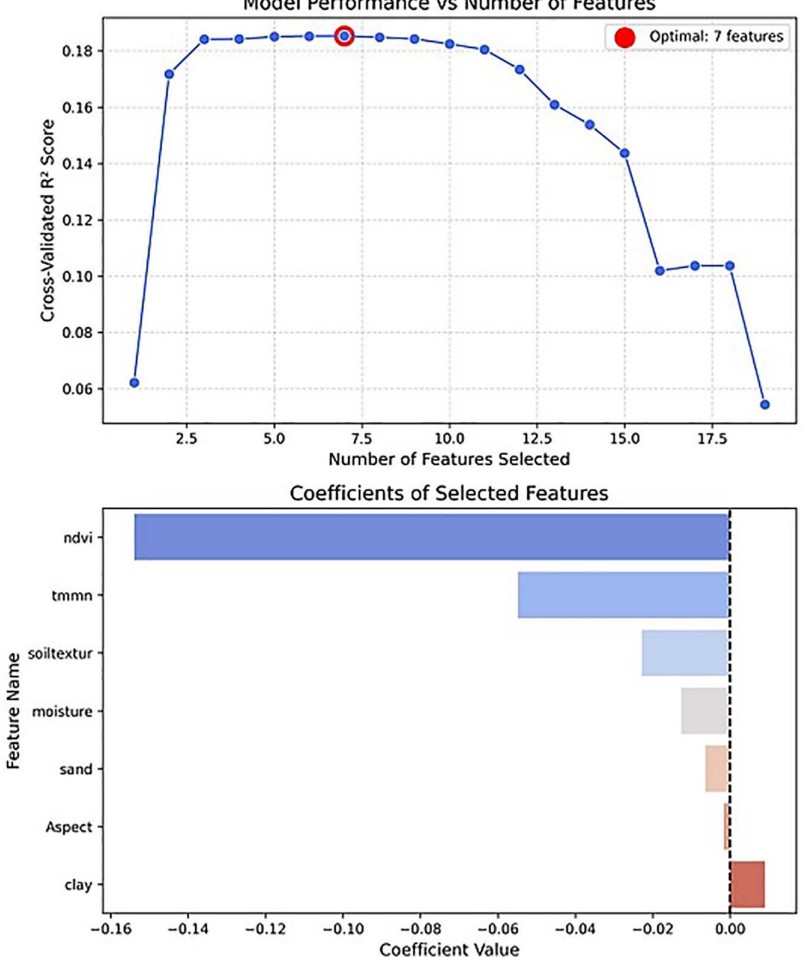

**Fig 9. R2-Score and Features Using the SFS Method, generated in Google Colab (https://colab.research.google.com).**

The Friedman test revealed that most models showed statistically significant differences in performance. Only the LSTM-OOA model, with a p-value of 0.4433, did not differ significantly from the base model. Meanwhile, the LSTM-HOA model, with the highest Chi-square statistic (116.05), exhibited the most substantial difference compared to the other models.

To evaluate the statistical stability of the proposed LSTM-WOA model, a 5-fold cross-validation repeated 15 times (total 75 runs) was conducted, as described in Section 2.5. The model demonstrated high stability with low coefficients of variation across all metrics: Accuracy (mean: 0.9336, CV: 0.31%), AUC (mean: 0.9479, CV: 0.40%), F1-Score (mean: 0.6479, CV: 2.56%), and Kappa (mean: 0.6113, CV: 2.93%). The narrow 95% confidence intervals further confirm the model's reliability for flood susceptibility mapping.

### 3.4. Evaluation of modeling accuracy of DLMs against ground truth

Regarding the models' power in flood detection based on Recall/Sensitivity, the OOA and WOA models demonstrated the best performance, respectively. This indicator reflects the models' ability to maximize the identification of flood-prone areas, which is particularly crucial for early warning systems (Table 9).

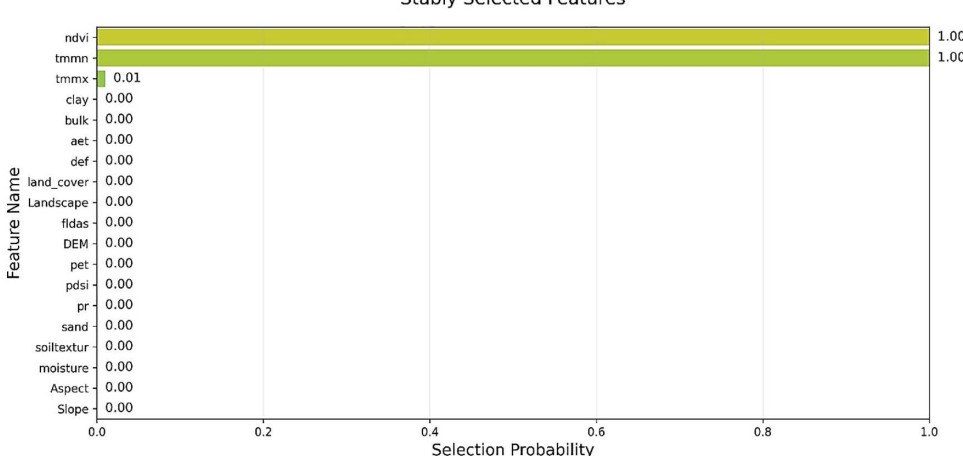

**Fig 10. Selection_Probability using the Stability Selection methods, generated in Google Colab (https://colab.research.google.com).**

In correctly predicting positive cases (i.e., accurately predicting flood-prone zones) based on the Precision metric, the WOA and OOA models were the top performers, respectively. This indicates that these models identified areas as flood-prone in approximately 60–61% of cases; in the remaining cases (~39%), the areas were not actually flood-prone.

In this context, the Specificity metric, which indicates the ability to correctly identify the absence of flood-prone areas, must be considered. The mentioned models (OOA, WOA) perform poorly on this metric. The best performance in this regard belongs to the CSA model, which excels at correctly identifying areas that are not flood-prone.

Considering the estimates from the previous two metrics, the best approach is to refer to the model's overall accuracy and consider the best balance between precision and recall (the F1-Score). Based on this, the WOA and OOA models are the top performers, respectively.

Given these interpretations and considering the need for a suitable balance between identifying flood zones and the reliability of predictions, the OOA model demonstrates the best overall performance. This model offers the best balance between flood detection power and overall reliability. It achieves the highest rate of flood zone detection among all models and also has the highest value for correct predictions.

## 4. Discussion

This study successfully developed and validated a novel hybrid framework for FSM by integrating a consensus-driven feature selection strategy with a metaheuristic-optimized LSTM model. The unanimous selection of NDVI by all nine feature selection methods confirms its role as a master variable for FSM in arid plains, as it is a direct proxy for Surface moisture retention, infiltration capacity, and antecedent wetness conditions [3,33]. The strong consensus on TMMN and Pr aligns with fundamental hydrological principles, where low nighttime temperatures reduce Evapo-transpiration and increase Soil moisture, and Pr is the primary flood trigger. The dominance of these climatic and biophysical variables over static topographic factors like Slope and Aspect is logical, given the extremely flat terrain of the Mesopotamian Plain in Khuzestan. This outcome demonstrates the ability of the framework to filter out irrelevant variables based on regional physiography, thereby reducing overfitting, which is a common issue in geospatial modeling where all available factors are often used indiscriminately. Compared to previous studies in the region that used a broader, less refined set of inputs [33], our consensus approach provides a more parsimonious and physically interpretable feature set.

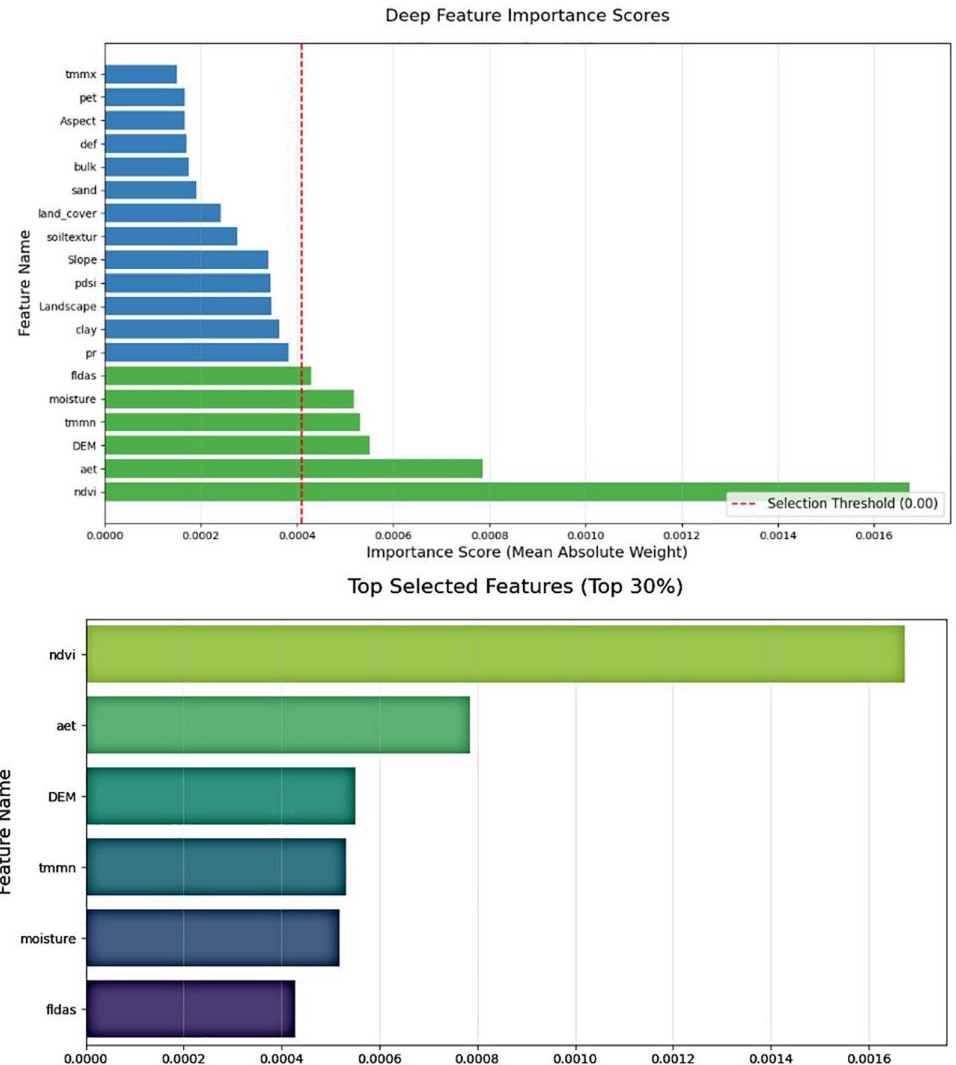

**Fig 11. Feature selection using Deep learning with ANN, generated in Google Colab (https://colab.research.google.com).**

The performance boost from metaheuristic optimization underscores the necessity of automated hyperparameter tuning for complex DLMs such as LSTM. Our comparative benchmark extends prior work that utilized single optimizers such as GWO or WOA [15,16], by systematically evaluating five algorithms, including novel ones such as OOA and HOA [18,19]. The superior and balanced performance of the LSTM-WOA model can be attributed to its unique bubble-net foraging mechanism [26]. This strategy balances exploration using random encircling to search the global hyperparameter space and exploitation using spiral updating to finely tune parameters around promising solutions. This dual capability is adept at navigating the high-dimensional, non-convex loss landscape of an LSTM, efficiently optimizing sensitive parameters such as the number of hidden units and learning rate to achieve an optimal bias-variance trade-off. In contrast, algorithms such as CSA, which excelled in Precision but had lower Recall, may exhibit a more conservative search pattern, potentially converging to safer but suboptimal local minima [30]. This comparative analysis guides algorithm selection in hydrological DLMs.

**Table 6. Comparative Table of Feature Selection Methods.**

| Method | Number of Selected Variables | Key Variables (Order of Importance) |
|---|---|---|
| Boruta | 10 | Pr, TMMN, NDVI, PDSI, Def, Soil moisture, TMMX, FLDAS, Aet, DEM |
| Boruta-SHAP | 10 | NDVI, Pr, DEM, Soil moisture, Aet, TMMN, Def, FLDAS, PDSI, TMMX |
| Elastic-Net | 9 | NDVI, TMMX, Aet, TMMN, Pet, Pr, Soil moisture, FLDAS, Soil texture |
| Mutual Information | 10 | NDVI, Def, Pet, Soil moisture, Aet, TMMN, FLDAS, TMMX, Pr, PDSI |
| Permutation Importance | 10 | NDVI, Def, Pr, Sand, Clay, DEM, TMMX, Aet, Soil moisture, TMMN |
| RFE | 9 | Pr, TMMN, NDVI, PDSI, Def, Soil moisture, FLDAS, TMMX, Aet |
| SFS | 7 | NDVI, TMMN, Soil texture, Soil moisture, Sand, Aspect, Clay |
| Stability Selection | 3 | NDVI, TMMN, TMMX |
| ANN | 7 | NDVI, Aet, DEM, TMMN, Soil moisture, FLDAS, Pr |

**Table 7. Comparative table of feature selection methods.**

| Variable | Number of Variable Selection Across 9 Methods | Big Picture | relative importance |
|---|---|---|---|
| NDVI` | ☑ 9/9 | Definitively Selected | ☆☆☆☆☆ |
| TMMN | ☑ 9/9 | Definitively Selected | ☆☆☆☆☆ |
| Pr | ☑ 7/9 | Strongly Selected | ☆☆☆☆☆ |
| Soil moisture | ☑ 7/9 | Strongly Selected | ☆☆☆☆☆ |
| Aet | ☑ 6/9 | Moderately Selected | ☆☆☆☆☆ |
| Def | ☑ 6/9 | Moderately Selected | ☆☆☆★☆ |
| TMMX | ☑ 6/9 | Moderately Selected | ☆☆☆★☆ |
| FLDAS | ☑ 6/9 | Moderately Selected | ☆☆☆★☆ |
| PDSI | ☑ 5/9 | Marginally Selected | ☆☆☆☆☆ |
| DEM | ☑ 4/9 | Weakly Selected | ☆☆★☆☆ |
| Pet | ☑ 3/9 | Weakly Selected | ☆☆☆☆☆ |
| Soil texture | ☑ 3/9 | Weakly Selected | ☆☆☆☆☆ |
| Sand | ☑ 3/9 | Weakly Selected | ☆☆☆☆☆ |
| Clay | ☑ 3/9 | Weakly Selected | ☆☆☆☆☆ |
| Slope | ✕ 1/9 | Eliminated | ☆☆☆☆☆ |
| Aspect | ✕ 1/9 | Eliminated | ☆☆☆☆☆ |
| Bulk | ✕ 1/9 | Eliminated | ☆☆☆☆☆ |
| Land Cover | ✕ 1/9 | Eliminated | ☆☆☆☆☆ |
| Landscape | ✕ 0/9 | Definitively Eliminated | ☆☆☆☆☆ |

The resultant susceptibility map, identifying high-susceptible zones in the northwestern and central plains, is hydrologically coherent. These areas correspond to the confluence zones of major rivers (Karun, Karkheh), regions with historical flood inundation, and areas with intensive agriculture that modifies natural drainage. This spatial validation aligns the model's output with real-world flood occurrences and offers actionable intelligence for planners. It advances beyond traditional susceptibility maps by providing a product born from a highly optimized and validated AI framework, thereby increasing decision-makers' confidence in using it for infrastructure prioritization, land-use zoning, and early warning systems, as called for in recent studies on disaster risk reduction [34].

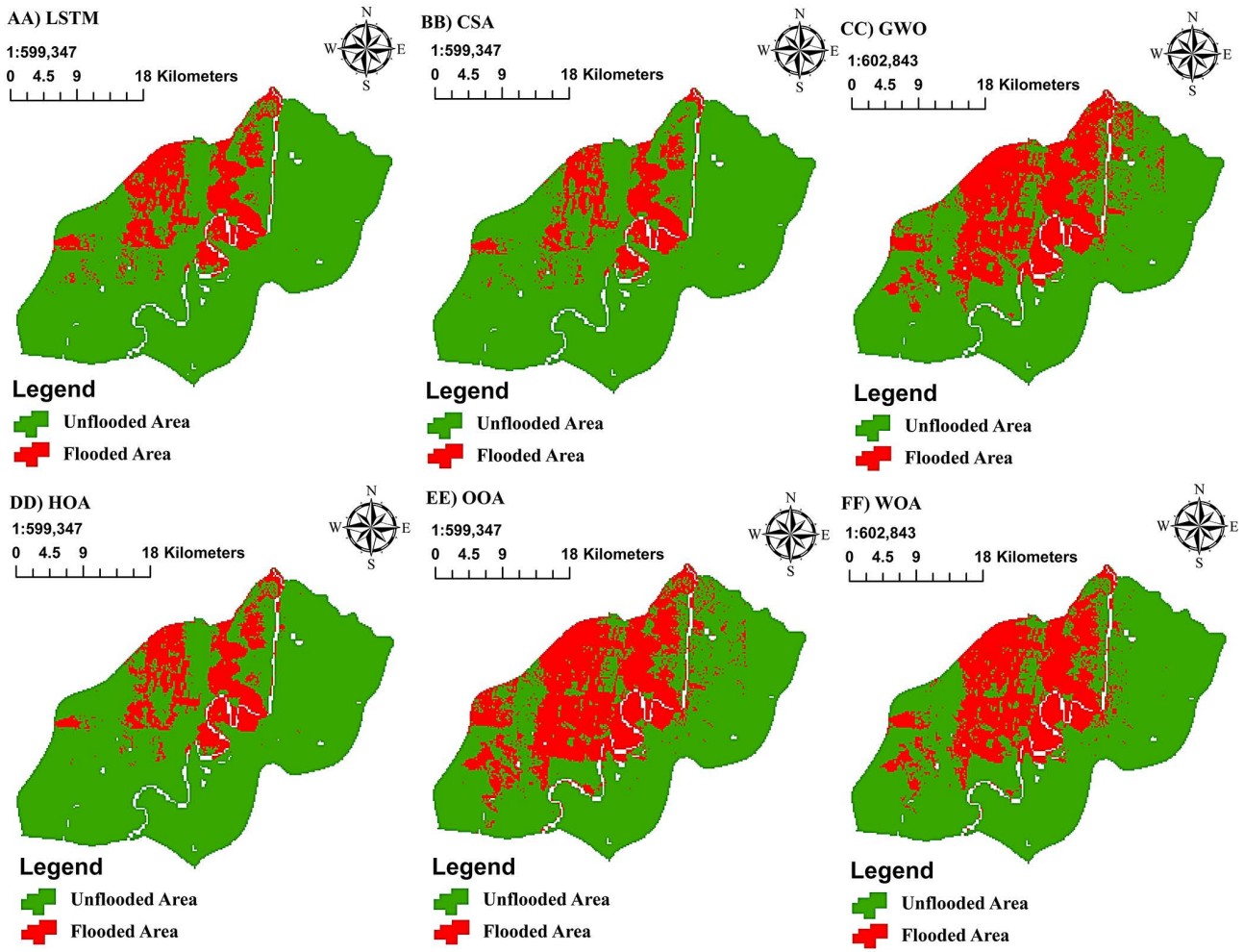

**Fig. 12. FSM Using the LSTM Model and Its Metaheuristic Optimization Procedures, generated in Google Colab (https://colab.research.google.com) and QGIS software (https://qgis.org).**

This study acknowledges several limitations that provide a clear agenda for future research. First, the reliance on static or temporally aggregated predictors may fail to capture the dynamic triggers of flash floods. Second, despite rigorous validation, the complex LSTM-metaheuristic model carries a risk of overfitting to the local conditions of Khuzestan, potentially limiting its transferability. Third, the framework's accuracy is contingent on the quality, consistency, and timeliness of the input GEE data, where errors or delays could hinder real-world applications. Fourth, additional blocking-based spatial cross-validation revealed that while the model maintained acceptable discriminative power (AUC > 0.82), classification metrics such as F1-score declined to approximately 0.50. This reduction highlights the model's sensitivity to spatial non-stationarity and class imbalance, confirming that random splitting may overestimate performance due to spatial autocorrelation. However, given the study's primary objective of producing the most accurate interpolation-based susceptibility map within the study area (27,000 dense points), random splitting remains appropriate. Future work should therefore prioritize: (1) integrating high-resolution, near-real-time data streams to model flood triggers more accurately; (2) conducting comprehensive transferability tests in diverse terrains to assess and improve generalizability; (3) developing robust uncertainty quantification methods to communicate prediction

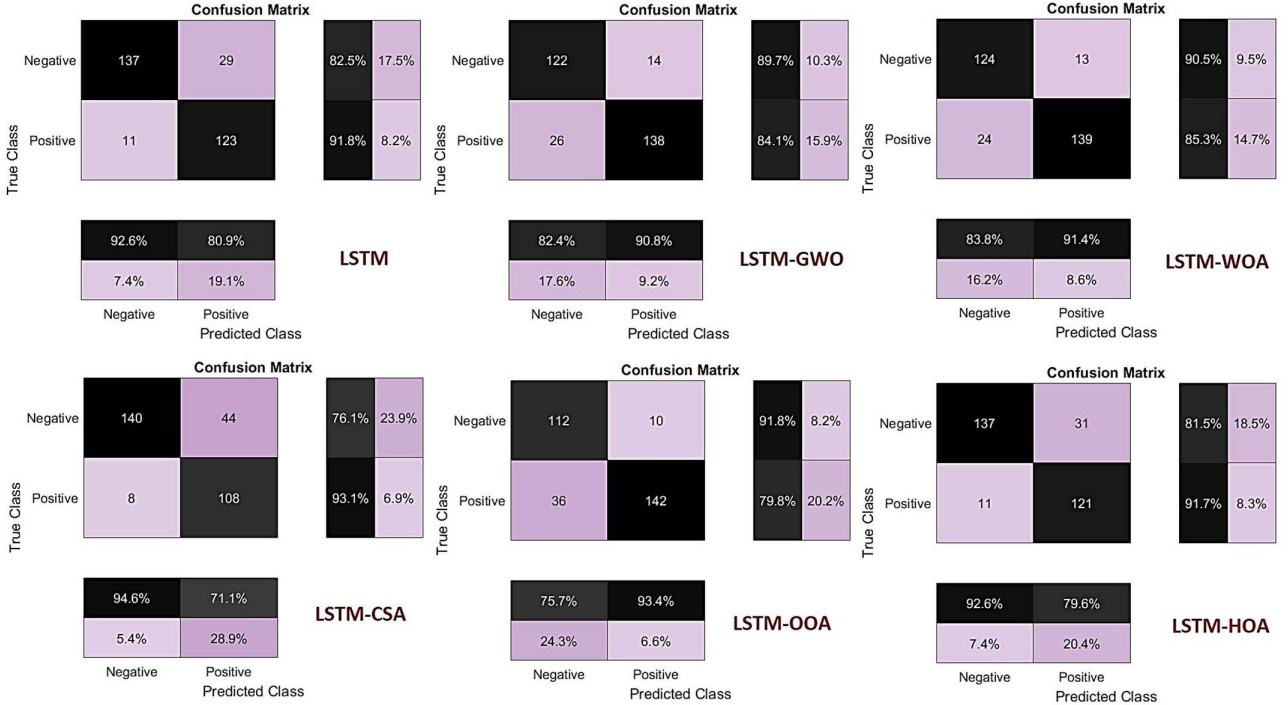

**Fig 13. Confusion Matrix for the LSTM Model and Its Metaheuristic Optimization Procedures, generated in Google Colab (https://colab.research.google.com).**

confidence for improved decision-making; and (4) addressing spatial non-stationarity through advanced spatially explicit modeling techniques.This discussion has interpreted the core findings, contextualized them within the existing literature, explained the mechanistic reasons for the success of dominant variables and the WOA optimizer, and candidly addressed the model's limitations. The study bridges critical gaps by offering a replicable, integrated pipeline that moves from robust feature selection to comparative model optimization, providing a significant step forward in creating reliable, interpretable, and practical tools for flood risk management in data-rich environments. Future work should focus on testing the framework's transferability to diverse terrains, integrating real-time data streams, and exploring adaptive metaheuristics for dynamic model tuning.

## 5. Conclusion

This study developed and rigorously validated a novel, integrated framework for high-resolution FSM by synergizing ensemble feature selection, metaheuristic optimization, and deep learning. The framework was successfully applied to the flood-prone Khuzestan Province, Iran, using multi-source data from GEE. The application of a strict majority-vote consensus rule across nine diverse feature selection algorithms definitively identified the NDVI and TMMN as the two most critical and stable predictors of FSM in the arid plains of Khuzestan. This finding underscores the primary role of surface moisture retention capacity (via vegetation) and antecedent soil moisture conditions (modulated by nighttime temperature) in driving flood genesis in this environment, providing a hydrologically interpretable and parsimonious feature set. Comparative hyperparameter optimization using five metaheuristic algorithms demonstrated that tuning significantly enhances LSTM performance. The LSTM model optimized with the LSTM-WOA emerged as the most balanced and superior model. It achieved the highest F1-Score (0.88) and Cohen's

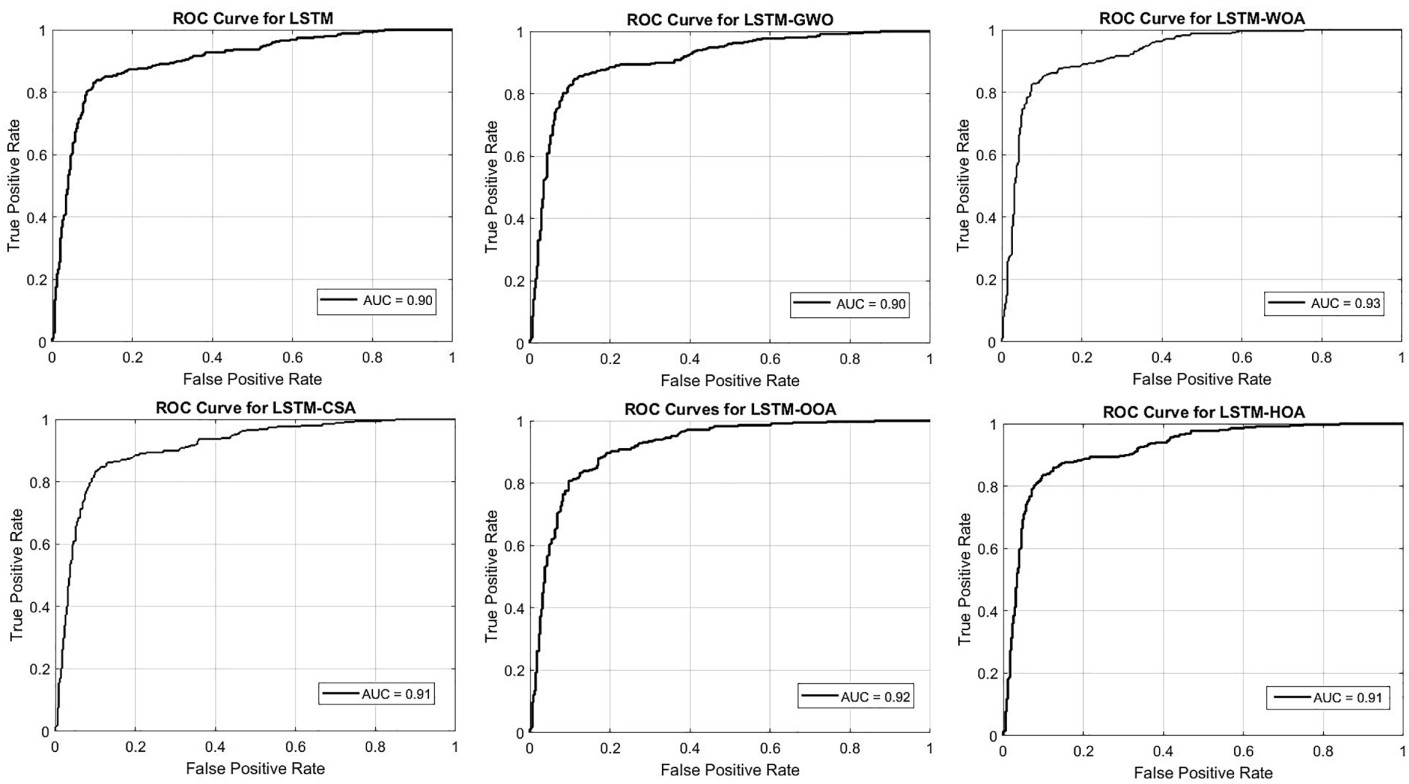

**Fig 14. ROC Curve for the LSTM Model and Its Metaheuristic Optimization Procedures, generated in Google Colab (https://colab.research.google.com).**

Kappa (0.75), indicating an optimal trade-off between precision and recall, and the strongest agreement with reality beyond chance. This confirms the effectiveness of WOA's bubble-net search strategy for navigating the complex hyperparameter space of deep learning models in geospatial applications. The proposed frequency-based consensus approach effectively resolved conflicts arising from different feature selection methods, moving beyond ad-hoc or single-method selections. It produced a robust, non-redundant set of 14 key variables, thereby reducing model input uncertainty and enhancing the generalizability and interpretability of the final susceptibility map. The final FSM generated by the optimal LSTM-WOA model accurately delineates the northwestern and central regions of Khuzestan as zones of very high susceptibility. This spatial pattern is hydrologically coherent, aligning with known floodplains, historical inundation records, and areas of concentrated agricultural activity. The map provides a reliable, data-driven foundation for land-use planners and disaster managers. Beyond the specific case study, this research contributes a replicable methodological pipeline that integrates robust data screening (consensus feature selection) with advanced model calibration (comparative metaheuristic optimization). This dual-layer optimization framework addresses critical gaps in the standard application of deep learning for FSM and can be adapted to other regions and environmental hazard modeling contexts. In summary, this study transcends conventional modeling by offering both a practical tool and a significant methodological advance. Future work should focus on integrating real-time data streams, testing the framework's transferability to diverse climatic and topographic settings, and incorporating socio-economic vulnerability layers to progress from susceptibility to comprehensive risk assessment.

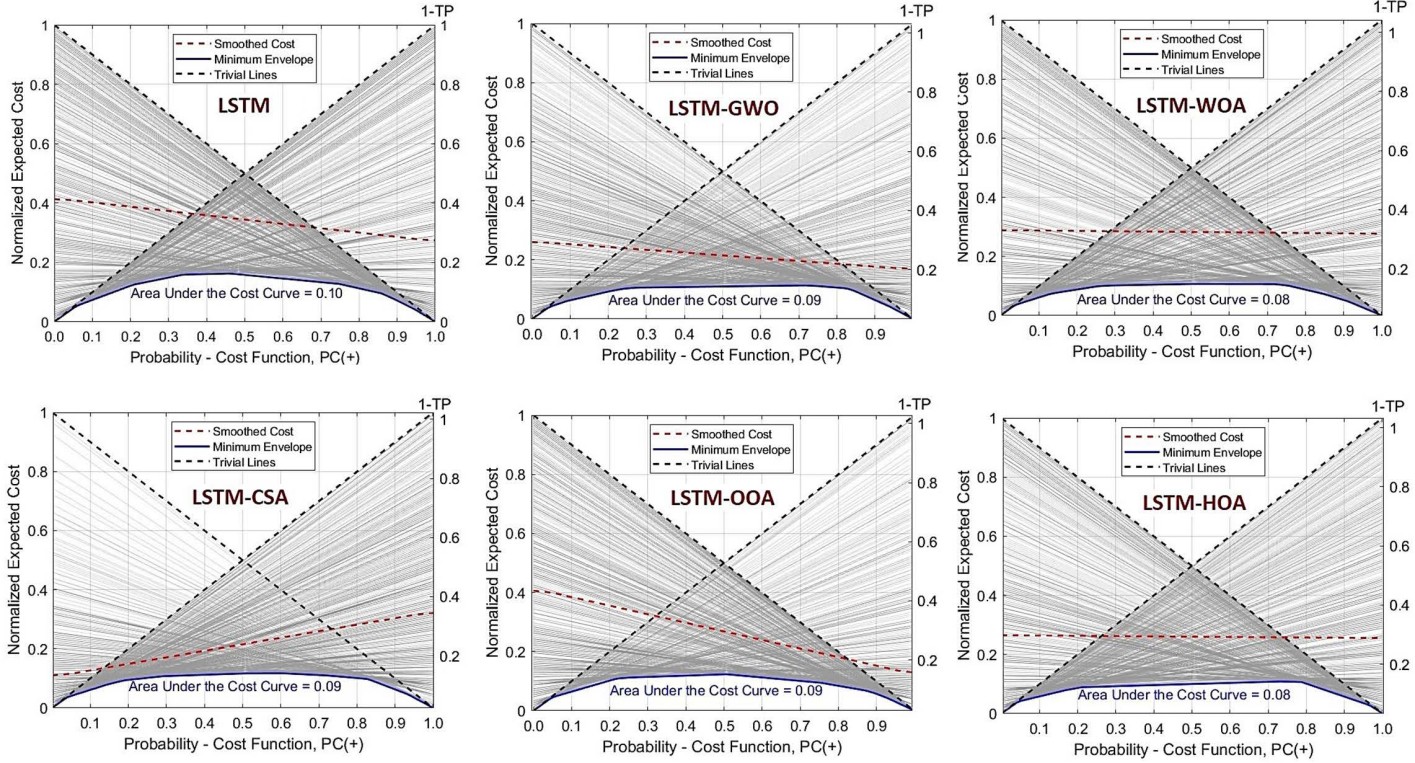

**Fig 15. Cost Curve for the LSTM Model and Its Metaheuristic Optimization Procedures, generated by the authors in Google Colab (https://colab.research.google.com).**

**Table 8. The Priority of the Evaluation Metric for the LSTM Model and Its Metaheuristic Optimization Procedures.**

| Evaluation category | Assessment criteria | LSTM | LSTM-GWO | LSTM-WOA | LSTM-CSA | LSTM-OOA | LSTM-HOA |
|---|---|---|---|---|---|---|---|
| Classification metrics | Precision | 0.92 | 0.84 | 0.85 | 0.93 | 0.80 | 0.92 |
| | Recall | 0.81 | 0.91 | 0.91 | 0.71 | 0.93 | 0.80 |
| | F1-Score | 0.86 | 0.87 | 0.88 | 0.81 | 0.86 | 0.85 |
| | Cohen's Kappa | 0.73 | 0.73 | 0.75 | 0.65 | 0.69 | 0.72 |
| Regression metrics | R | 0.75 | 0.77 | 0.78 | 0.76 | 0.76 | 0.78 |
| | $R^2$ | 0.51 | 0.58 | 0.57 | 0.50 | 0.54 | 0.53 |
| | MAE | 0.26 | 0.22 | 0.26 | 0.25 | 0.27 | 0.26 |
| | RMSE | 0.35 | 0.32 | 0.33 | 0.35 | 0.34 | 0.34 |
| Statistical tests | Fridman test p-value | 1.9046e-21 | 5.695e-05 | 2.5314e-15 | 3.0488e-07 | 0.44 | 6.321e-26 |
| | Chi-sq | 95.42 | 19.55 | 67.22 | 30.01 | 1.63 | 116.05 |

**Table 9. Evaluation of the Modeling Accuracy of DLMs Against the Ground Truth.**

| Metric | CSA | GWO | HOA | LSTM | OOA | WOA |
|---|---|---|---|---|---|---|
| Sample Count (N) | 300 | 300 | 300 | 300 | 300 | 300 |
| True Positive (TP) | 37 | 80 | 61 | 52 | 88 | 85 |
| True Negative (TN) | 47 | 20 | 26 | 32 | 16 | 19 |
| False Positive (FP) | 26 | 53 | 38 | 41 | 57 | 54 |
| False Negative (FN) | 11 | 5 | 16 | 25 | 3 | 6 |
| Overall Accuracy | 69.42% | 82.64% | 71.90% | 69.42% | 85.95% | 85.95% |
| Precision | 58.73% | 60.15% | 61.62% | 55.91% | 60.69% | 61.15% |
| Recall (Sensitivity) | 77.08% | 94.12% | 79.22% | 67.53% | 96.70% | 93.41% |
| F1-Score | 66.67% | 73.39% | 69.32% | 61.18% | 74.58% | 73.91% |
| Specificity | 64.38% | 27.40% | 40.63% | 43.84% | 21.92% | 26.03% |

## Supporting information

**S1 File. Detailed information regarding the implementation codes of the metaheuristic algorithms, sensitivity analysis of key hyperparameters, and the complete input dataset is provided in Supporting Information File S1.** (RAR)

## Acknowledgments

We thank DeepSeek for editing the grammar of the text.

## Author contributions

**Conceptualization:** Mohammad Kazemi.

**Methodology:** Reza Naderi Samani.

**Project administration:** Mohammad Kazemi.

**Software:** Reza Naderi Samani.

**Supervision:** Mohammad Kazemi.

**Validation:** Reza Naderi Samani.

**Visualization:** Narges Kariminejad.

**Writing – original draft:** Narges Kariminejad.

**Writing – review & editing:** Narges Kariminejad.

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
