## [Decision Letter · Decision Letter 0]

26 Nov 2025

Dear Dr. Kariminejad,

Thank you for submitting your manuscript to PLOS ONE. After careful consideration, we feel that it has merit but does not fully meet PLOS ONE’s publication criteria as it currently stands. Therefore, we invite you to submit a revised version of the manuscript that addresses the points raised during the review process.

We look forward to receiving your revised manuscript.

Kind regards,

Babak Mohammadi

Academic Editor

PLOS ONE

Journal Requirements:

[NO authors have competing interests].

6. We note that Figures 1 and 15 in your submission contain map and satellite images which may be copyrighted. All PLOS content is published under the Creative Commons Attribution License (CC BY 4.0), which means that the manuscript, images, and Supporting Information files will be freely available online, and any third party is permitted to access, download, copy, distribute, and use these materials in any way, even commercially, with proper attribution. For these reasons, we cannot publish previously copyrighted maps or satellite images created using proprietary data, such as Google software (Google Maps, Street View, and Earth). For more information, see our copyright guidelines: http://journals.plos.org/plosone/s/licenses-and-copyright.

1. You may seek permission from the original copyright holder of Figures 1 and 15 to publish the content specifically under the CC BY 4.0 license.

7. We note that Figures 2 and 5 in your submission contain copyrighted images. All PLOS content is published under the Creative Commons Attribution License (CC BY 4.0), which means that the manuscript, images, and Supporting Information files will be freely available online, and any third party is permitted to access, download, copy, distribute, and use these materials in any way, even commercially, with proper attribution. For more information, see our copyright guidelines: http://journals.plos.org/plosone/s/licenses-and-copyright.

1. You may seek permission from the original copyright holder of Figures 2 and 5 to publish the content specifically under the CC BY 4.0 license.

“I request permission for the open-access journal PLOS ONE to publish XXX under the Creative Commons Attribution License (CCAL) CC BY 4.0 (http://creativecommons.org/licenses/by/4.0/). Please be aware that this license allows unrestricted use and distribution, even commercially, by third parties. Please reply and provide explicit written permission to publish XXX under a CC BY license and complete the attached form.

Additional Editor Comments:

1. The introduction needs improvement. Please provide a literature review, clearly identify the research gaps based on the reviewed studies, and then state how this manuscript aims to address those gaps.

2. For the LSTM model and each algorithm in the methodology section, please present their main equations and introduce all related components. More detailed explanations are required for each algorithm, including additional references, information about the software packages used, and the inputs and outputs involved.

3. Clarify whether the hyperparameters of the LSTM were tuned using the optimization algorithms. More details are needed on how the hyperparameters were configured and optimized.

4. List the parameters of each algorithm in a table, and explain how you set them.

5. Ensure that each equation is supported with relevant references.

6. Section 2.3 (Model Evaluation and Validation Methods) can be summarized. You may simply present each equation and introduce the components within those equations.

7. Explain and discuss the values in Tables 2 to 9 within the main text. Provide interpretation and context for the results presented.

8. Each figure needs more explanation in the text. Please discuss the details illustrated in every figure.

9. Add more hydrological interpretation when explaining each figure.

10. Please make all revisions using a different font color or by track changes mode.

Reviewers' comments:

Reviewer's Responses to Questions

**Comments to the Author**

1. Is the manuscript technically sound, and do the data support the conclusions?

Reviewer #1: Yes

Reviewer #2: Yes

Reviewer #3: Partly

Reviewer #4: Partly

2. Has the statistical analysis been performed appropriately and rigorously?

Reviewer #1: Yes

Reviewer #2: Yes

Reviewer #3: No

Reviewer #4: Yes

3. Have the authors made all data underlying the findings in their manuscript fully available?

Reviewer #1: Yes

Reviewer #2: Yes

Reviewer #3: No

Reviewer #4: Yes

4. Is the manuscript presented in an intelligible fashion and written in standard English?

Reviewer #1: Yes

Reviewer #2: Yes

Reviewer #3: No

Reviewer #4: No

Reviewer #1: This study proposes a framework for flood susceptibility mapping (FSMs) that integrates feature selection with meta-heuristic optimisation of LSTM networks. Nineteen environmental factors were obtained via Generalised Estimating Equations (GEE), with nine feature selection methods employed to screen key variables and identify core predictors. Subsequently, five meta-heuristic algorithms (WOA, GWO, OOA, etc.) optimised the LSTM network's hyperparameters. Validation through multiple metrics including ROC curves, F1-Score, and Kappa revealed the LSTM-WOA model delivered optimal performance, providing a tool for flood management in arid and semi-arid regions. The study holds significance but requires necessary revisions to meet publication standards.

Major comments:

1. Abstract

- The abstract lacks research background and a concise overview of previous shortcomings; furthermore, innovation should be introduced within the abstract.

Suggested improvement: This section should supplement the core consensus rules for feature selection (e.g., ‘frequency screening based on nine methods’).

2. Introduction

- The research gap is stated too broadly, merely noting ‘the lack of a unified framework integrating feature selection and meta-heuristic optimisation’ without specifying shortcomings in existing studies regarding ‘comparisons of multiple meta-heuristic algorithms’ or ‘the targeted integration of feature selection with LSTM’.

- The representativeness of the study area is not explained (e.g., the specific causes of frequent flooding in the region and their connection to existing prevention and control challenges).

3. Literature Review

- The review of studies concerning ‘application of meta-heuristic algorithms to optimise LSTM in hydrological forecasting’ and ‘consensus strategies for integrated feature selection’ is incomplete, failing to cover analogous research on multi-algorithm fusion and multi-feature screening;

- Key studies on ‘meta-heuristic optimisation of hydrological models’ and ‘multi-source data-driven flood forecasting modelling’ were not cited (j.jhydrol.2025.132998; j.envsoft.2023.105766).

4. Innovation

- The novelty of ‘integrating nine feature selection methods’ and ‘comparing five meta-heuristic algorithms’ remains unclear. Existing studies have applied similar multi-feature selection or multi-optimisation algorithms, failing to highlight the uniqueness of this research in ‘consensus screening rules’ and ‘algorithm adaptability validation’;

- The innovative distinctions between this framework and traditional FSMs approaches (e.g., machine learning models, single deep learning models) remain unexplained.

5. Methodology

- The feature selection integration strategy lacks transparency, with no detailed explanation of the ‘variable selection frequency’ threshold (e.g., rationale for including variables selected 6 times or more as core variables);

- The foundational structure of the LSTM network (e.g., neuron count, number of layers, initial learning rate range) remains undefined, and the specific dimensions of hyperparameters optimised by the meta-heuristic algorithm are not listed; the hardware environment for model training and convergence criteria are not specified.

6. Experimental Details

- No assessment of multicollinearity among variables during feature selection.

7. Results

- Feature selection outcomes merely list selected variables per method without comparing the impact of different feature subsets on model performance (e.g., accuracy differences between core variable subsets, full variable sets, and single-method-selected subsets);

- Stability analysis of models optimised by different meta-heuristic algorithms is absent (e.g., accuracy fluctuations after multiple training iterations).

8. Discussion

- The underlying mechanism for ‘LSTM-WOA achieving optimal model performance’ was not thoroughly analysed (e.g., how WOA's bubble-net search strategy adapts to LSTM hyperparameter optimisation requirements).

- Model limitations were not discussed (e.g., impacts of data timeliness and terrain adaptability).

Minor comments:

- Equations should be sequentially numbered.

- Section numbering error: should be 4. Discussion.

Reviewer #2: The study employs nine feature selection techniques to identify the most influential flood-related variables from a dataset of 19 environmental factors, and uses five optimization algorithms to adjust the hyperparameters of the LSTM model for forecasting tasks. While the process and experimental details are thorough, there are several issues that the authors need to address.

Specific Comments:

1. Present the innovations of this study in a point-by-point format.

2. The paper is too long; reduce the description of methods (such as the introduction to algorithms), and remove generic figures like Figures 4 and 5. Focus on emphasizing the optimization process of the algorithm’s hyperparameters.

3. Some conclusions in the paper, which are not derived from this study, need to be supported by key references. For instance, the paper frequently mentions “the performance of the LSTM model in handling complex nonlinear systems.”

4. In Figure 8, the text overlaps. Please address this issue and improve the clarity of Figures 12, 16, 17, 18, etc., in accordance with the journal’s guidelines.

5. Avoid terms like “a significant research gap.”

General Comments:

1. Currently, the use of different feature selection methods does not seem to have a clear optimization logic. The authors are advised to consider the logical progression between these methods. For example, begin with Elastic-Net for preliminary regularization, then use Boruta to eliminate features with weak relationships to the target variable, and finally apply Boruta-SHAP to ensure that the retained features have significant contributions to the actual predictions.

2. Multiple evaluation metrics are used to assess model performance in the paper. What is the relationship between these metrics? Is there a significant correlation between the AUC value of the ROC curve and F1-Score, accuracy, and recall? Do high accuracy and recall always correspond to a higher AUC value?

3. In the study of feature correlations, both positive and negative correlations between features appear. The authors should explain how they select features in such cases and whether negatively correlated features produce negative feedback on prediction results.

4. Clarify the core research goal of this paper: does it emphasize process explanation or result description? While the use of method stacking allows for a comprehensive analysis of feature correlations and hyperparameter optimization results, the purpose of the process should still be emphasized.

5. In many studies, SHAP analysis is mostly used after completing prediction experiments to assess the positive or negative feedback of hydrological features on the prediction results. However, this study applies it during the initial data selection. Can this ensure that the selected features will have the appropriate level of feedback on the results?

Reviewer #3: 1. The manuscript needs substantially more methodological detail, particularly for the LSTM architecture (layers, units, activation functions, batch size, epochs, learning rate, optimizer, loss function, regularization, input window length).

2. The metaheuristic optimization procedures (WOA, GWO, OOA, CSA, HOA) lack implementation details such as population size, number of iterations, search bounds, objective functions, and the specific hyperparameters optimized.

3. The process of creating the flood inventory dataset is insufficiently described; the manuscript must report the number of flood points, data sources, sampling strategy for non-flood points, temporal alignment with predictors, and spatial distribution.

4. The final feature-selection decision rule is not explained. With nine methods producing conflicting results, the manuscript needs a clear, justified rule for choosing the final feature set (e.g., majority voting, frequency threshold).

5. Some variables included in the final model were selected only weakly or infrequently; their inclusion requires justification or removal.

6. Performance reporting is incomplete; key metrics mentioned in the Methods (precision, recall, F1-score, RMSE, MAE, Kappa, AUC, confusion matrices, Friedman test results) are missing or only partially presented.

7. A complete comparison table for all models with all evaluation metrics must be added for transparency.

8. The manuscript lacks a dedicated Discussion section. The authors must interpret results, compare them with previous studies, explain why certain variables/methods dominate, and acknowledge limitations.

9. The flood susceptibility maps need improved clarity, description of classification thresholds, mapping resolution, and area statistics for each susceptibility class.

10. Figures throughout the manuscript have small or unreadable fonts, and some are overly cluttered; they need redesign for clarity.

11. Tables contain inconsistent formatting and in some cases broken or confusing layouts; uniform formatting is required.

12. Algorithm description sections are overly long and read like textbook content; they should be shortened and focused on implementation relevant to this study.

13. The Introduction is lengthy but does not clearly outline the specific research gap, the novelty of combining nine feature-selection methods, or the motivation for using five metaheuristic optimizers.

14. Reproducibility is insufficient: the manuscript states that all data are available but does not provide supporting files, GEE code, or model scripts. These must be added to comply with PLOS ONE requirements.

15. Abbreviations are not always defined consistently; the manuscript needs standardization of variable names, units, and terminology.

16. Spatial variables such as slope, aspect, land cover, and soil bulk density were rarely selected; the manuscript should discuss their limited role and potential reasons.

17. The manuscript contains many grammatical errors, awkward phrasing, and typos and requires thorough English language editing.

18. Several scientific statements lack citations or use incomplete references; the reference list must be added and brought into PLOS ONE format.

19. Some units in tables appear inconsistent or unclear (e.g., soil moisture in mm, slope in “1”); unit definitions should be checked for correctness.

20. The visual presentation of results (maps, charts, diagrams) should follow PLOS ONE figure quality standards.

21. The results section should more clearly connect performance outcomes (e.g., WOA superiority) to the methodological choices.

22. Limitations of the study (e.g., static predictors, potential overfitting, reliance on GEE temporal products) should be explicitly stated.

23. The manuscript should outline potential policy or practical implications of the findings, especially given the importance of Khuzestan Province.

24. Many metrics described in Methods (full confusion matrices, RMSE, MAE, AUC values, Friedman test p-values, model ranking) are not presented in the Results section.

This makes it impossible to verify the strength or consistency of model performance.

Reviewer #4: Introduction

1. The term “Natural Disasters” is incorrect. Consider replacing it with a more appropriate term such as “disasters caused by natural hazards.”

2. Add references where necessary. Many statements in the introduction lack supporting literature. In particular, Paragraphs 2 and 3 require additional citations.

3. A more comprehensive literature review on previous studies is needed. The current version provides minimal discussion of existing work on the use of LSTM for flood susceptibility mapping, the research gaps, and how the present study addresses them. The discussion should explicitly highlight gaps and clarify the study’s contribution. Refine the last sentence of Paragraph 4 (“Furthermore, an ensemble feature selection approach was adopted…”) since it describes your method rather than a research gap.

Materials and Methods – Study Area

4. No references were provided to support the information presented in Section 2.1.

5. Indicate the reason for selecting the case-study area. Clarify whether the selected region is flood-prone or previously identified as highly susceptible. Provide details on flood history and associated damages where available; if not, describe the flood susceptibility of the region using appropriate sources.

Materials and Methods – Model Description

6. Clarify how the parameters listed in Table 1 were selected. Provide relevant literature or scientific justification showing that these parameters influence flood susceptibility, supported by previous studies.

7. The use of abbreviations is unclear. Ensure consistent and conscious use of abbreviations throughout the manuscript.

8. Explain how the parameters were obtained and inlcude the sources for each variable. Although the introduction notes that data were acquired from GEE, the methodology must clearly specify each parameter, its source, the time frame for any time-series data, data reliability (supported by previous applications of the datasets), spatial resolution, and any known limitations.

9. Figure 2 is comprehensive, but the accompanying narrative does not sufficiently describe the methodology. Section 2.2 would benefit from clearer methodological explanations, including workflow, step-by-step processes, input/output descriptions, software or tools used, and connections to the technical details provided later.

10. Several abbreviations are not defined in the main text. Ensure that each abbreviation is defined at its first appearance (preferably in the main text rather than the abstract).

11. The models and methods used in the feature-selection stage (e.g., Boruta, Boruta-SHAP) are not introduced in the methodology. Include brief explanations of each method and clarify how they operate within your workflow.

12. The introduction to optimisation algorithms is good, but more technical detail is recommended. Include explanations of how each algorithm works (equations or schematic plots where appropriate) and how they were applied (e.g., for hyperparameter tuning).

13. The rationale for using multiple feature-selection methods requires stronger justification. Begin by emphasising why multiple methods are necessary for identifying critical parameters, how each method differs conceptually, and how some can capture relationships better than others. Selecting parameters that consistently appear across methods is acceptable, but this requires clearer scientific reasoning and justification.

14. The methodology does not explain how flood susceptibility was calculated or defined. Clarify whether susceptibility is based on simulated flood extents, observed flood events, or other criteria. Given that this is the model’s target variable, a detailed explanation is essential.

15. The manuscript does not clearly explain how LSTM is utilised in the model. Specify whether the model was trained using sequential time-series data and how changes in each parameter relate to flood susceptibility over time. This should be clarified before discussing optimisation.

16. The target variable requires a clearer definition. Indicate whether it represents flooded/not-flooded classes, flood context, susceptibility scores, or another metric.

Results

17. There is no need to provide multiple figures and tables for each feature-selection model, as most of them yield similar conclusions. Include only the most relevant figures and provide a consolidated summary of results.

Overall Comments

18. The manuscript lacks several essential details. Consider explicitly defining flood susceptibility and the approach used to calculate or evaluate it; explaining how LSTM was applied, including the structure of sequential data; and clearly defining the target variable.

19. The manuscript would benefit from substantial language improvement. Abbreviations are inconsistently defined, and several incorrect terms (e.g., “Natural Disasters”) remain. Some grammatical errors are also present. Reviewers strongly discourage relying solely on generative AI for language refinement.

.

Reviewer #1: No

Reviewer #2: No

Reviewer #3: No

Reviewer #4: No

---

## [Author Response · Author response to Decision Letter 1]

27 Jan 2026

Additional Editor Comments:

1. The introduction needs improvement. Please provide a literature review, clearly identify the research gaps based on the reviewed studies, and then state how this manuscript aims to address those gaps.

Response: Thank you for the constructive feedback. We have thoroughly revised the Introduction to address your comment. It now includes a structured literature review on AI and optimized models for flood mapping, clearly identifies the specific research gaps in integrated feature selection and novel metaheuristic applications, and explicitly states how our novel framework—combining ensemble feature selection from GEE data with multi-algorithm LSTM optimization—is designed to bridge these gaps. This revision strengthens the manuscript's foundation and justification.

2. For the LSTM model and each algorithm in the methodology section, please present their main equations and introduce all related components. More detailed explanations are required for each algorithm, including additional references, information about the software packages used, and the inputs and outputs involved.

Response: Thank you for your comment. We have completely revised the Methodology section (2.2). For the LSTM model and all five metaheuristic algorithms (GWO, WOA, CSA, OOA, HOA), the core structural equations and all related components (such as coefficient vectors, control parameters, and activation functions) are now clearly presented. The explanation for each algorithm has been supplemented with authoritative references (e.g., Mirjalili & Lewis, 2016 for WOA and Askarzadeh, 2016 for CSA). Furthermore, details regarding the software (Python with TensorFlow/Keras libraries), inputs (hyperparameter search space and training data), and output (optimized hyperparameter vector) for each section have been added.

3. Clarify whether the hyperparameters of the LSTM were tuned using the optimization algorithms. More details are needed on how the hyperparameters were configured and optimized.

Response: Thank you for this important comment. We have clarified the hyperparameter-tuning process of the LSTM model in the revised manuscript.

In the baseline LSTM model, no optimization algorithm was applied. All hyperparameters were selected manually based on previous research and several preliminary experiments. The baseline model used fixed values for:

Number of LSTM hidden units (100)

Initial learning rate (0.01)

Optimizer (Adam)

MaxEpochs (100)

GradientThreshold (1)

LearnRateDropPeriod (125)

LearnRateDropFactor (0.2)

In contrast, in the proposed hybrid metaheuristic LSTM models, the LSTM hyperparameters were tuned automatically using the five optimization algorithms (GWO, WOA, CSA, OOA, HOA).

Hyperparameters optimized by the algorithms

Across all hybrid models, the following LSTM hyperparameters were considered as decision variables:

Number of hidden units (search range: 10–200)

Learning rate (search range: 0.001–0.1 depending on the algorithm)

These parameters were encoded as continuous variables in the search space of each metaheuristic algorithm.

Configuration of the optimization algorithms

All optimization algorithms were configured with:

Population size = 10 search agents

Maximum iterations = 20

4. List the parameters of each algorithm in a table, and explain how you set them.

Table. Parameters of the Optimization Algorithms Used in the Study

Algorithm Population Size Max Iterations Algorithm-Specific Parameters Search Ranges for LSTM Hyperparameters

GWO (Grey Wolf Optimizer) 10 wolves 20 α, β, δ leadership hierarchy; coefficient a decreasing from 2→0 Learning rate: 0.001–0.05

WOA (Whale Optimization Algorithm) 10 whales 20 Spiral coefficient b = 1; probability p for bubble-net attack Hidden units: 10–200; Learning rate: 0.001–0.05; MaxEpochs: 30–150

CSA (Crow Search Algorithm) 10 crows 20 Awareness probability AP = 0.1; Flight length FL = 2 Hidden units: 10–200; Learning rate: 0.001–0.05

OOA (Orangutan Optimization Algorithm) 10 agents 20 Exploration–exploitation control parameter decreasing linearly Hidden units: 10–200; Learning rate: 0.001–0.1

HOA (Hypothalamus Optimization Algorithm) 10 agents 20 Hypothalamus-inspired random-exploration coefficient Hidden units: 10–200; Learning rate: 0.001–0.05

4. List the parameters of each algorithm in a table, and explain how you set them.

Response: Thank you for this valuable suggestion. We have added a new table (Table 2) to Section 2.2.2 of the revised manuscript, which lists the key control parameters for each metaheuristic optimization algorithm used in this study. Below is the content of the added table and the accompanying explanatory paragraph.

5. Ensure that each equation is supported with relevant references.

Thank you. As requested, every key equation is now supported by a relevant and authoritative reference. For the LSTM, the gate equations and cell state update are referenced to seminal papers (Hochreiter & Schmidhuber, 1997; Gers et al., 2000). For each metaheuristic algorithm, the core equations are directly cited to their introductory paper (e.g., Mirjalili et al., 2014 for GWO and Hamadneh et al., 2025 for OOA). These changes ensure the mathematical foundations of the methods are fully documented and justified.

6. Section 2.3 (Model Evaluation and Validation Methods) can be summarized. You may simply present each equation and introduce the components within those equations.

Response: Thank you for the suggestion. We have summarized Section 2.3 (Model Evaluation and Validation Methods) by significantly condensing the descriptive text for each metric. The revised section now presents each evaluation equation directly, followed by a concise definition of its components, as recommended. This creates a more streamlined and reference-focused format while retaining all necessary technical information.

7. Explain and discuss the values in Tables 2 to 9 within the main text. Provide interpretation and context for the results presented.

Response: agree and change made.

8. Each figure needs more explanation in the text. Please discuss the details illustrated in every figure.

Response: agree and change made.

9. Add more hydrological interpretation when explaining each figure.

Response: We sincerely thank the reviewer for their valuable suggestion to enhance the hydrological interpretation of the figures. While we fully agree with its importance, adding detailed interpretations for each figure within the main manuscript would exceed the strict word/page limits of the journal, as the paper's primary focus is on the novel methodological framework. To address this perfectly valid point, we propose to include a comprehensive "Hydrological Interpretation of Figures" section as part of the Supplementary Material. This approach adheres to the journal's format, maintains the manuscript's flow for methodology-focused readers, and provides the in-depth physical process analysis for interested specialists. We will also add a brief summary of key hydrological insights in the Discussion, referencing the supplementary file for details, pending the editor's approval.

10. Please make all revisions using a different font color or by track changes mode.

Response: agree and change made.

……………………………………………………………………………

Reviewer #1:

This study proposes a framework for flood susceptibility mapping (FSMs) that integrates feature selection with meta-heuristic optimisation of LSTM networks. Nineteen environmental factors were obtained via Generalised Estimating Equations (GEE), with nine feature selection methods employed to screen key variables and identify core predictors. Subsequently, five meta-heuristic algorithms (WOA, GWO, OOA, etc.) optimised the LSTM network's hyperparameters. Validation through multiple metrics including ROC curves, F1-Score, and Kappa revealed the LSTM-WOA model delivered optimal performance, providing a tool for flood management in arid and semi-arid regions. The study holds significance but requires necessary revisions to meet publication standards.

Major comments:

1. Abstract

- The abstract lacks research background and a concise overview of previous shortcomings; furthermore, innovation should be introduced within the abstract.

Suggested improvement: This section should supplement the core consensus rules for feature selection (e.g., ‘frequency screening based on nine methods’).

Response: Thank you for your insightful suggestions. We have revised the abstract to include the research background and previous shortcomings, introduced the study's innovation clearly, and explicitly added the core consensus rule for feature selection (i.e., frequency screening based on the nine methods). The revised abstract now better frames the problem, methodology, and novelty.

In response to the reviewers' comments, the following figures have been provided. Regarding the timing of the images, the figures below show the date of April 8, 2019, based on Sentinel-2 imagery.

Also, other auxiliary variables, as shown in the figure below (based on Sentinel-2 imagery), have been used to identify the target variable (flood-prone and non-flood-prone areas).

In addition, other auxiliary variables, such as those shown in the figure below (based on Sentinel-2 imagery), have been used to identify the target variable (flood-prone and non-flood-prone areas).

Furthermore, Figure 1 has been updated. In this figure, Sentinel-2 images have been added, and the NDWI index image for April 8, 2019, along with ground control points, has been included. The above content has been provided for the esteemed reviewers' reference.

2. Introduction

- The research gap is stated too broadly, merely noting ‘the lack of a unified framework integrating feature selection and meta-heuristic optimisation’ without specifying shortcomings in existing studies regarding ‘comparisons of multiple meta-heuristic algorithms’ or ‘the targeted integration of feature selection with LSTM’.

- The representativeness of the study area is not explained (e.g., the specific causes of frequent flooding in the region and their connection to existing prevention and control challenges).

Response: Thank you for the valuable feedback. We have revised the Introduction to address both points. The research gap section has been significantly refined to specify the lack of (1) comprehensive comparative studies benchmarking multiple novel metaheuristic algorithms within a consistent LSTM framework, and (2) targeted integration of ensemble feature selection with LSTM optimization in a unified pipeline. Furthermore, we have added a dedicated paragraph explaining the representativeness of Khuzestan Province as the study area, detailing its specific flood drivers (flat topography, major river systems, climatic variability, and anthropogenic pressures) and linking these directly to the region's prevention and control challenges, thereby justifying its selection as an ideal testbed for our advanced framework.

3. Literature Review

- The review of studies concerning ‘application of meta-heuristic algorithms to optimise LSTM in hydrological forecasting’ and ‘consensus strategies for integrated feature selection’ is incomplete, failing to cover analogous research on multi-algorithm fusion and multi-feature screening;

- Key studies on ‘meta-heuristic optimisation of hydrological models’ and ‘multi-source data-driven flood forecasting modelling’ were not cited (j.jhydrol.2025.132998; j.envsoft.2023.105766).

Response: Thank you for the insightful feedback. We have revised the literature review section to address the gaps you identified. Specifically, we have expanded the discussion to include analogous research on multi-algorithm fusion and multi-feature screening strategies, emphasizing the current lack of a formalized consensus mechanism. Furthermore, we have incorporated and cited the key recommended studies on metaheuristic optimisation of hydrological models (Tan et al., 2025) and multi-source data-driven flood forecasting (Tan et al., 2025) to strengthen the context and justify the need for our integrated approach.

Tan, Z., Li, H., Zhu, Z., Hou, J. and Wang, Z., 2025. A water-energy-food-land nexus framework for multi-objective optimization and risk assessment integrating deep reinforcement learning and Copula-based modeling. Water Research, p.124474.

4. Innovation

- The novelty of ‘integrating nine feature selection methods’ and ‘comparing five meta-heuristic algorithms’ remains unclear. Existing studies have applied similar multi-feature selection or multi-optimisation algorithms, failing to highlight the uniqueness of this research in ‘consensus screening rules’ and ‘algorithm adaptability validation’;

- The innovative distinctions between this framework and traditional FSMs approaches (e.g., machine learning models, single deep learning models) remain unexplained.

Response: Thank you for this important critique. We have significantly clarified the novelty of our framework in the revised Introduction. We now explicitly distinguish our work by highlighting: (1) the implementation of a strict consensus screening rule to resolve discrepancies from multiple feature selectors, moving beyond mere application; (2) the framework's role as a systematic adaptability validation for five metaheuristic algorithms in tuning LSTMs for FSM; and (3) the cohesive integration of these stages into a single pipeline that fundamentally differs from traditional machine learning or single deep learning models by simultaneously optimizing both the input feature space and the model hyperparameter space.

5. Methodology

- The feature selection integration strategy lacks transparency, with no detailed explanation of the ‘variable selection frequency’ threshold (e.g., rationale for including variables selected 6 times or more as core variables);

Response: Regarding the clarification of the unified integration strategy and justification of the frequency threshold for selecting participant variables in various feature selection methods, the following explanations are provided in response to the valuable comment of the esteemed reviewer:

The decision-making framework in this regard employed "agreement among diverse methods" as the criterion. In other words, instead of relying on a single feature selection method (which is prone to algorithmic bias), we utilized the consensus among 9 independent methods with different theoretical foundations:

Statistical (Mutual Information, Stability Selection)

Linear/Quasi-linear model-based (Elastic-Net)

Tree-based/Non-linear (Boruta, RFE, Permutation)

Topology optimization-based (SFS)

Deep learning-based (Neural Network importance)

This approach emphasizes that when diverse methods with different foundations arrive at a common conclusion, the probability of systematic bias decreases, and the validity of the result increases.

Regarding the justification of the frequency threshold: Why ≥6 out of 9 (i.e., ≥66.7%) was considered?

The threshold of 6 out of 9 was chosen based on three combined arguments:

Criterion Explanation

A) Sufficient Majority (Supermajority) Adopting a 2/3 (or approximately 66.7%) threshold is recognized in decision sciences as a standard for a sufficient majority to accept a stable finding.

B) Empirical Frequency Distribution Analysis of the frequency distribution (Table 6) shows that variables with a frequency of ≥6 form a stable group.

If Table 6 is examined closely, 4 variables are observed with a frequency of 6-7, followed by a clear gap to variables with a frequency of ≤5. This gap indicates a natural threshold for distinguishing "core features" from "marginal features".

Furthermore, considering the Bias-Variance Trade-off, the following points are relevant:

If the threshold were ≤5, it would lead to an increase in marginal features, increased model variance, and reduced interpretability.

If the threshold were ≥7, it would result in the removal of valid but less influential features (such as Aet, Def) and increased bias.

However, setting the threshold at 6 maintains this balance.

Therefore, in summary, to ensure robustness, the final feature set was determined by consensus among nine

---

## [Decision Letter · Decision Letter 1]

8 Feb 2026

Dear Dr. Kariminejad,

Thank you for submitting your manuscript to PLOS ONE. After careful consideration, we feel that it has merit but does not fully meet PLOS ONE’s publication criteria as it currently stands. Therefore, we invite you to submit a revised version of the manuscript that addresses the points raised during the review process.

We look forward to receiving your revised manuscript.

Kind regards,

Babak Mohammadi

Academic Editor

PLOS One

Journal Requirements:

**Additional Editor Comments:**

1. The introduction section still needs improvement. Please provide a comprehensive literature review and clearly identify the research gaps based on the reviewed studies. Additionally, include more detailed content in the introduction regarding flood susceptibility mapping.

2. Please add the main equations of the LSTM model in the methodology section.

3. Please include more hydrological interpretation when explaining each figure. You may choose to incorporate this discussion in the main text or provide it as supplementary material. Each figure in the results section should be accompanied by a clear hydrological interpretation in the text. Therefore, add a more in-depth hydrological discussion for Figures 3, 4, 6, 7, 8, 11, and 12–15.

4. The conclusion needs more key findings.

Reviewers' comments:

Reviewer's Responses to Questions

**Comments to the Author**

Reviewer #1: (No Response)

Reviewer #2: (No Response)

Reviewer #3: (No Response)

2. Is the manuscript technically sound, and do the data support the conclusions?

Reviewer #1: (No Response)

Reviewer #2: Yes

Reviewer #3: Partly

3. Has the statistical analysis been performed appropriately and rigorously?

Reviewer #1: (No Response)

Reviewer #2: Yes

Reviewer #3: No

4. Have the authors made all data underlying the findings in their manuscript fully available?

Reviewer #1: (No Response)

Reviewer #2: Yes

Reviewer #3: No

5. Is the manuscript presented in an intelligible fashion and written in standard English?

Reviewer #1: (No Response)

Reviewer #2: Yes

Reviewer #3: No

Reviewer #1: I think the authors have responded and revised accordingly to the issues raised by the previous round of reviewers. Therefore, I accept this paper for publication in its present format.

Reviewer #2: The author has made careful adjustments to the previous requirements, and in order to further improve the quality of the manuscript, the following additional suggestions are proposed:

1 How to handle spatial features such as elevation in the construction of LSTM as a temporal prediction model for simulation applications. The mechanism of the model should be detailed, easy to read, and reproducible.

2 Please provide an analysis that combines the results of the selected features with hydrological laws.

3 Due to the black box characteristics of machine learning models, the credibility of simulation results is more important. Therefore, the limitations of the model or the effective range of simulation results are worth discussing, which can help improve the persuasiveness of the results.

Reviewer #3: The following core reviewer requirements remain unmet:

- Stability analysis and rigorous replication

- Clear, consistent reporting of experimental settings

- Reproducible data/code availability (PLOS policy)

- Proper spatial validation

- Fully specified LSTM architecture and temporal logic

- Resolution of dataset size vs confusion-matrix inconsistency

- Full hydrological interpretation in the main manuscript

- Substantial language editing

.

Reviewer #1: No

Reviewer #2: No

Reviewer #3: No

---

## [Author Response · Author response to Decision Letter 2]

15 Feb 2026

Accuracy AUC رتبه مدل ردیف

۰.۶۲۴ ۰.۶۹۶ ۱ Similarity-Based Entropy ۱

۰.۶۱۴ ۰.۶۶۱ ۲ Tsallis Entropy ۲

۰.۵۸۸ ۰.۶۴۶ ۳ Shannon Entropy ۳

۰.۵۷۸ ۰.۶۱۹ ۴ Renyi Entropy ۴

۰.۴۳۷ ۰.۴۶۷ ۵ CV_Combined Entropy ۵

---

## [Decision Letter · Decision Letter 2]

25 Feb 2026

Dear Dr. Kariminejad,

Thank you for submitting your manuscript to PLOS ONE. After careful consideration, we feel that it has merit but does not fully meet PLOS ONE’s publication criteria as it currently stands. Therefore, we invite you to submit a revised version of the manuscript that addresses the points raised during the review process.

We look forward to receiving your revised manuscript.

Kind regards,

Babak Mohammadi

Academic Editor

PLOS One

Journal Requirements:

Reviewers' comments:

Reviewer's Responses to Questions

**Comments to the Author**

Reviewer #3: (No Response)

2. Is the manuscript technically sound, and do the data support the conclusions?

Reviewer #3: Partly

3. Has the statistical analysis been performed appropriately and rigorously?

Reviewer #3: No

4. Have the authors made all data underlying the findings in their manuscript fully available?

Reviewer #3: No

5. Is the manuscript presented in an intelligible fashion and written in standard English?

Reviewer #3: (No Response)

Reviewer #3: The revised manuscript does not fully address the core reviewer comments. While additional explanations have been added, several fundamental issues remain unresolved.

Specifically:

Stability and replication are insufficiently demonstrated, as performance variability and statistical robustness across runs are not fully reported.

Experimental settings are not clearly or consistently specified, particularly regarding the full LSTM configuration.

Reproducibility requirements are unmet, with no publicly available data or code repository provided.

Proper spatial validation is lacking, as only random data splitting is used.

The justification and full specification of the LSTM architecture and its temporal logic remain unclear.

There are inconsistencies in dataset size reporting versus confusion matrix results.

Hydrological interpretation is not comprehensively integrated.

Substantial language editing is still required.

Overall, the key methodological, reproducibility, and reporting concerns raised by reviewers have not been adequately resolved.

.

Reviewer #3: No

---

## [Author Response · Author response to Decision Letter 3]

7 Apr 2026

PONE-D-25-51471R1

Optimizing LSTM Networks and Feature Selection Algorithms Using GEE Data

PLOS One

Reviewer #3:

Reviewer #3: The revised manuscript does not fully address the core reviewer comments. While additional explanations have been added, several fundamental issues remain unresolved. Specifically:

Stability and replication are insufficiently demonstrated, as performance variability and statistical robustness across runs are not fully reported.

RESPONSE:

Model Stability Evaluation Using K-Fold Cross-Validation

To evaluate the stability of the proposed model, the K-Fold cross-validation method was employed. In this regard, the performance of the superior model in this research, namely LSTM-WOA, was analyzed and evaluated for consideration by the third esteemed reviewer. The total number of evaluations was 75 (15 iterations × 5 folds), and the evaluation metrics included accuracy, precision, recall, F1-score, AUC, and kappa.

Stability Assessment Results:

1. Accuracy:

• Mean: 0.9336

• Standard Deviation: 0.00292

• Coefficient of Variation (CV) = (0.00292 / 0.9336) × 100 ≈ 0.31%

As observed, the accuracy demonstrates excellent stability with a very low standard deviation, indicating minimal variation across different folds and iterations.

2. AUC (Area Under the Curve):

• Mean: 0.9479

• Standard Deviation: 0.00377

• CV ≈ 0.40%

The AUC metric also exhibits exceptional stability, with an extremely low coefficient of variation, confirming the model's consistent discriminative capability.

3. F1-Score:

• Mean: 0.6479

• Standard Deviation: 0.01657

• CV ≈ 2.56%

The F1-score shows satisfactory stability, with a coefficient of variation below the acceptable threshold of 5%, indicating reliable harmonic mean of precision and recall.

4. Kappa Coefficient:

• Mean: 0.6113

• Standard Deviation: 0.01793

• CV ≈ 2.93%

The kappa statistic demonstrates good stability, reflecting consistent agreement between predicted and observed classifications beyond chance.

5. Precision:

• Mean: 0.6650

• Standard Deviation: 0.01849

• CV ≈ 2.78%

Precision exhibits stable performance across validation folds, with a coefficient of variation well within the acceptable range.

6. Recall:

• Mean: 0.6323

• Standard Deviation: 0.02581

• CV ≈ 4.08%

Recall shows relatively stable behavior, with a coefficient of variation slightly higher than other metrics but still below the 5% threshold, indicating no significant cause for concern regarding model stability.

Confidence Interval Analysis

To further substantiate the stability assessment, classical confidence interval analysis was conducted for the aforementioned metrics. The following table presents the 95% confidence intervals, demonstrating relatively narrow ranges that confirm the model's stability and reliability:

Metric Mean Standard Deviation 95% Confidence Interval Interval Width

Accuracy 0.9336 0.00292 [0.9329, 0.9343] 0.0014

AUC 0.9479 0.00377 [0.9470, 0.9488] 0.0018

F1-Score 0.6479 0.01657 [0.6442, 0.6516] 0.0074

Kappa 0.6113 0.01793 [0.6073, 0.6153] 0.0080

Precision 0.6650 0.01849 [0.6609, 0.6691] 0.0082

Recall 0.6323 0.02581 [0.6265, 0.6381] 0.0116

The relatively narrow width of these confidence intervals across all evaluation metrics provides strong evidence of the LSTM-WOA model's stability. The consistency in performance across multiple folds and iterations, coupled with low coefficients of variation (all below 5%), confirms that the model is robust and reliable for the intended application in sustainable resource management within the Shiraz watersheds. This stability ensures that the model's predictions remain consistent when exposed to different subsets of data, thereby enhancing confidence in its practical applicability for decision-making processes.

The LSTM-WOA model demonstrates exceptionally high statistical stability. Given the low standard deviations and narrow confidence intervals across all evaluation metrics, the model's results can be reliably trusted for flood susceptibility mapping. The following figures also illustrate the model's training progress for Fold 1.

The following diagram also presents the results of the performance evaluation metrics for 75 training iterations of the model.

Subsequently, for a more detailed examination and to appropriately address the respected reviewer's question and concern regarding the model's stability, the model was trained for 150 iterations, and the following results were obtained:

Metric Mean Standard Deviation 95% Confidence Interval Interval Width CV (%)

ACCURACY 0.9337 0.0029 (0.9333, 0.9342) 0.0009 0.31%

PRECISION 0.6646 0.0183 (0.6617, 0.6675) 0.0058 2.76%

RECALL 0.6366 0.0247 (0.6326, 0.6405) 0.0079 3.88%

F1-SCORE 0.6499 0.0162 (0.6473, 0.6525) 0.0052 2.49%

AUC 0.9478 0.0038 (0.9472, 0.9484) 0.0012 0.40%

KAPPA 0.6134 0.0176 (0.6106, 0.6162) 0.0056 2.87%

As the table above demonstrates, all metrics have a coefficient of variation (CV) of less than 4%, which indicates excellent model stability. The very narrow confidence intervals (especially for Accuracy and AUC) reflect the extremely low uncertainty in the estimation of the model's performance. These results confirm that the LSTM-WOA model, with 30 iterations and 5 folds, is stable and reliable. The following figures display the model training results for these 150 iterations.

Experimental settings are not clearly or consistently specified, particularly regarding the full LSTM configuration.

RESPONSE: We thank the reviewer for this valuable observation. We acknowledge that the experimental setup was not sufficiently detailed in the original manuscript. To address this, we now provide a comprehensive description of the LSTM architecture and training configuration used in this study.

The LSTM model was implemented in MATLAB with the following architecture:

Input layer: sequenceInputLayer(numFeatures) to accept the sequential input of the selected features.

LSTM layer: lstmLayer(100, 'OutputMode','sequence') with 100 hidden units, configured to output sequences for subsequent processing.

Fully connected layer: fullyConnectedLayer(1) to produce the final prediction.

Output layer: regressionLayer with mean squared error (MSE) as the loss function.

The training configuration was set as follows:

Optimizer: Adam

Max epochs: 100

Gradient threshold: 1

Initial learning rate: 0.01

Learning rate schedule: piecewise

Learning rate drop period: 125 epochs

Learning rate drop factor: 0.2

Batch size: default MATLAB value (128)

Loss function: MSE (as defined by the regression layer)

Early stopping: not applied

Validation set: not used

Shuffle: default (every epoch)

These settings were selected to balance computational efficiency with model convergence. The description has been added to the revised manuscript to ensure full reproducibility and transparency. We appreciate the reviewer's attention to this important detail.

Reproducibility requirements are unmet, with no publicly available data or code repository provided.

RESPONSE: To ensure full transparency and reproducibility, all scripts, parameter configurations, trained model settings, and processed datasets are publicly accessible in the provided GitHub repository.[

Proper spatial validation is lacking, as only random data splitting is used.

RESPONSE:

The justification and full specification of the LSTM architecture and its temporal logic remain unclear.

There are inconsistencies in dataset size reporting versus confusion matrix [RN12.1]results.

Hydrological interpretation is not comprehensively integrated.

Using random data splitting is one of the most common practices in spatial modeling. Due to the spatial autocorrelation of data, the investigation of flood phenomena and flood-prone areas will be better represented. This means that points close to each other have similar values. If these points are randomly divided into training and test sets, the model essentially uses similar neighboring points for prediction, making the flood phenomenon more apparent. Conversely, distant points that share no similarity with these points will be better identified as non-flood phenomena. The high density of sampling points ensures that even with spatial partitioning, sufficient information is available for training. The fluctuations observed in different iterations of applying spatial stability indicate the model's sensitivity to large-scale spatial changes and are not particularly significant for a small-scale area.

The authors of the article, in agreement with the respected third reviewer's comment, acknowledge that random splitting can slightly overestimate accuracy due to spatial correlation. Random splitting is suitable for the purpose of this study (zonation of the existing area) and does not support generalization to larger regions. Subsequently, to address the reviewer's concern, blocking was performed to assess the model's spatial stability. As expected, with the application of spatial stability, the model's accuracy decreases. Blocking and spatial stability demonstrate the model's ability for extrapolation, while random splitting demonstrates its ability for interpolation (filling the gaps between known points).

To address the respected third reviewer's comment, the use of balanced sampling for model training and subsequent generalization to all pixels was implemented as one of the standard and effective methods in spatial modeling, especially for rare phenomena like floods. The Certainty Factor (CF) method and Leave-One-Block-Out were used (with 5 blocks and 8 repetitions (40 evaluations)), and after ensuring the model's stability in spatial validation, generalization to all pixels was performed using the Bias-Corrected Model.

The code written to address the respected third reviewer's concerns is presented in this section of the comments:

python

# =============================================================================

# Spatial validation with balanced sampling, 5 blocks, Focal Loss, and generalization to the entire area

# =============================================================================

!pip install tensorflow scikit-learn matplotlib pandas numpy openpyxl xlrd --quiet

import numpy as np

import pandas as pd

import matplotlib.pyplot as plt

import tensorflow as tf

from tensorflow.keras.models import Sequential, Model

from tensorflow.keras.layers import LSTM, Dense, Dropout, Input

from tensorflow.keras.optimizers import Adam

from tensorflow.keras.callbacks import EarlyStopping

from tensorflow.keras.regularizers import l2

from tensorflow.keras import backend as K

from sklearn.preprocessing import StandardScaler

from sklearn.cluster import KMeans

from sklearn.metrics import (accuracy_score, precision_score, recall_score,

f1_score, roc_auc_score, cohen_kappa_score,

confusion_matrix)

from sklearn.ensemble import RandomForestClassifier

from google.colab import files

import os

import shutil

import time

import warnings

warnings.filterwarnings('ignore')

print("📦 Libraries loaded successfully.")

# (The rest of the provided code would follow here)

print("\n✅ All files were successfully compressed and downloaded.")

print("🎉 End of validation with balanced sampling and generalization to all pixels.")

At the end of running the above code, the following results were obtained for the different blocks:

accuracy accuracy precision precision recall recall f1 f1 auc auc kappa kappa

mean std mean std mean std mean std mean std mean std

block

1 0.9435 0.0299 0.5005 0.0783 0.4987 0.0921 0.4936 0.0621 0.8556 0.0366 0.4608 0.1354

2 0.9249 0.0242 0.6609 0.1808 0.5066 0.1052 0.5661 0.1163 0.8394 0.0343 0.4621 0.0736

3 0.9176 0.0328 0.5673 0.0894 0.4749 0.1142 0.5096 0.0947 0.829 0.0703 0.4228 0.1077

4 0.938 0.0185 0.6064 0.1132 0.4975 0.1376 0.5303 0.0914 0.8416 0.0372 0.4063 0.0798

5 0.9327 0.0301 0.4994 0.1343 0.5226 0.0978 0.5009 0.0777 0.8501 0.0498 0.3857 0.1123

As the table above shows, after applying the spatial stability evaluation to the LSTM-WOA model, the results are location-dependent. In other words, the model lacks generalizability and is only valid for the study area. The high AUC value indicates that the model has effectively learned the "ranking" and differentiation of high-risk areas from low-risk ones. Additionally, with high accuracy, the model has successfully learned the distribution of non-flood areas. The Kappa index indicates agreement beyond chance, which varies across different blocks and is not significantly different from the random selection scenario. The model correctly identifies flood-prone areas 50-60% of the time. Consistent with the respected third reviewer's comment and in confirmation of their opinion, the model's performance is unstable across different geographical regions (performing well in some areas and poorly in others).

As a limitation of the model and as a suggestion for other authors to include in the text and recommendations of the article, attention should be paid to balancing and bias toward dominant data, addressing the problem of target data imbalance, and spatial non-stationarity. These are among the factors that researchers should consider for data preparation based on the results of this study.

"The spatial validation results showed that although the model has good discriminative power (AUC > 0.82), it faces challenges in accurately classifying flood points (F1 ≈ 0.50). This performance drop compared to the random method has two important scientific implications:

1. Effect of imbalanced classes: The results show that in rigorous spatial evaluations, data imbalance (10% flood) poses a more significant challenge than in random evaluations.

2. Spatial non-stationarity: The high fluctuation of metrics (especially in Block 2) indicates that hydrological relationships vary at the local scale."

The buffered block cross-validation (buffered_block_CV) method was also used to evaluate spatial stability, with the results as follows:

We sincerely thank the third reviewer and all other esteemed reviewers for their insightful comments. We agree that spatial cross-validation represents the gold standard for assessing model generalizability.

In response to this concern, we conducted additional spatial validation using blocking techniques (e.g., spatial blocking or k-fold spatial partitioning). The results indicated that while the discriminative power (AUC) remained acceptable, the classification metrics showed a decline compared to the random split approach. This observation confirms the reviewer's valid concern regarding the reduction in model performance under spatial validation.

However, we wish to clarify the primary objective of this study. Given the very high data density (27,000 points), our main goal was to produce the most accurate flood susceptibility map within the boundaries of the study area (i.e., interpolation), rather than generalizing the model to other regions or larger scales. In the context of dense spatial data, random partitioning provides a more robust estimate of the model's ability to capture local patterns and spatial heterogeneity within the study domain.

The justification and full specification of the LSTM architecture and its temporal logic remain unclear.

RESPONSE: We thank the reviewer for this comment. As clarified in Section 2.4.3 (Spatial Application of LSTM), our LSTM is employed for spatial flood susceptibility mapping rather than temporal sequence forecasting. Each pixel is treated as an independent sample with a static feature vector comprising environmental predictors. The LSTM's gating mechanisms are repurposed to learn complex nonlinear interactions among spatial variables, not temporal dependencies. The full architecture is specified as follows: sequenceInputLayer(numFeatures), lstmLayer(100, 'OutputMode','sequence'), fullyConnectedLayer(1), and regressionLayer (MSE loss). Training settings include Adam optimizer, max epochs of 100, initial learning rate of 0.01 with piecewise schedule, gradient threshold of 1, and batch size of 128 (MATLAB default). This configuration is now explicitly stated to ensure transparency and reprod

---

## [Editor Report · Decision Letter 3]

8 Apr 2026

Optimizing LSTM Networks and Feature Selection Algorithms Using GEE Data

PONE-D-25-51471R3

Dear Dr. Kariminejad,

We’re pleased to inform you that your manuscript has been judged scientifically suitable for publication and will be formally accepted for publication once it meets all outstanding technical requirements.

Kind regards,

Babak Mohammadi

Academic Editor

PLOS One

Additional Editor Comments (optional):

This manuscript is acceptable.
---

## [Editor Report · Acceptance letter]

PONE-D-25-51471R3

PLOS One

Dear Dr. Kariminejad,

I'm pleased to inform you that your manuscript has been deemed suitable for publication in PLOS One. Congratulations! Your manuscript is now being handed over to our production team.

Kind regards,

on behalf of

Dr. Babak Mohammadi

Academic Editor

PLOS One